# Induction of open-form bile canaliculus formation by hepatocytes for evaluation of biliary drug excretion

Hiroshi Arakawa [1,4], Yuya Nakazono [1,4], Natsumi Matsuoka[1], Momoka Hayashi[1], Yoshiyuki Shirasaka[1], Atsushi Hirao [2,3] & Ikumi Tamai [1✉]

Biliary excretion is a major drug elimination pathway that affects their efficacy and safety. The currently available in vitro sandwich-cultured hepatocyte method is cumbersome because drugs accumulate in the closed bile canalicular lumen formed between hepatocytes and their amounts cannot be mealsured directly. This study proposes a hepatocyte culture model for the rapid evaluation of drug biliary excretion using permeation assays. When hepatocytes are cultured on a permeable support coated with the cell adhesion protein claudins, an open-form bile canalicular lumen is formed at the surface of the permeable support. Upon application to the basolateral (blood) side, drugs appear on the bile canalicular side. The biliary excretion clearance of several drugs, as estimated from the obtained permeabilities, correlates well with the reported in vivo biliary excretion clearance in humans. Thus, the established model is useful for applications in the efficient evaluation of biliary excretion during drug discovery and development.

[1] Faculty of Pharmaceutical Sciences, Institute of Medical, Pharmaceutical and Health Sciences, Kanazawa University, Kanazawa 920-1192, Japan. [2] Division of Molecular Genetics, Cancer Research Institute, Kanazawa University, Kanazawa 920-1192, Japan. [3] WPI Nano Life Science Institute (WPI-Nano LSI), Kanazawa University, Kanazawa 920-1192, Japan. [4] These authors contributed equally: Hiroshi Arakawa, Yuya Nakazono. ✉email: tamai@p.kanazawa-u.ac.jp

Pharmacokinetic characteristics determine the efficacy, safety, and sometimes the dosage forms of drugs. Therefore, adequate pharmacokinetic studies are essential to increase the success rate during drug discovery and development. Hepatic drug handling, including metabolism and biliary excretion via drug-metabolizing enzymes and transporters, is often critical because the characteristics of the hepatobiliary disposition are linked to drug safety issues. Accordingly, several in vitro methods to characterize hepatic drug disposition have been used in preclinical studies, such as primary cultures of cryopreserved human hepatocytes, isolated human microsomes, transporter-gene-transfected cells, liver-derived cultured cells, and hepatic microphysiological systems (MPS)[1–3].

In the past three decades, the prediction of hepatic clearance has improved in cases dominated by cytochrome P450 (CYP)-mediated phase I drug metabolism. Since drugs subject to CYP metabolism potentially exhibit a risk of drug-drug interaction (DDI) and hepatotoxicity by forming reactive metabolites, recent drug designs tend to avoid metabolism by CYPs. As a result, there is an increasing number of compounds with hepatic clearance via biliary excretion of the parent and phase II enzyme-mediated conjugates. Furthermore, small molecules targeting protein-protein interactions (PPIs) have attracted the attention of researchers[4]. A PPI interface of approximately 1500–3000 Å is larger than the receptor-ligand binding pocket (300–1000 Å), leading to a larger molecular weight of small molecules for PPI (>400 Da) compared with traditional ones (200–500 Da)[5]. However, it is well known that an increase in the molecular weight of compounds is associated with increased biliary excretion[6]. Accordingly, the importance of evaluating the biliary excretion characteristics of drugs is increasing, while in vitro methods to evaluate this process are limited.

Sandwich-cultured hepatocytes (SCHs) are in vitro models currently used to assess the biliary excretion potential of drugs by obtaining the so-called biliary excretion index (BEI)[7]. They have been used in many hepatic drug disposition studies[8–10]. In SCHs, closed bile canalicular lumens are formed via tight junctions between hepatocytes. Biliary excretion is then evaluated by measuring the accumulated drugs in the closed lumens as obtained from the difference in the total amount of drug associated with the cells in the presence and absence of $Ca^{2+}/Mg^{2+}$ used to manipulate the tight junctions. Accordingly, although SCHs retain the physiological function of hepatocytes, uptake assays using SCHs are insufficient for analysing time-dependent dynamic hepatic disposition of drugs, large-scale screening, and compounds with limited biliary excretion.

In contrast to SCHs, permeation assays are easily evaluated via transcellular transport measurement using cell monolayers cultured on permeable supports, such as Caco-2 cells. This method is popular for rapidly screening membrane permeability in industrial and academic laboratories. Accordingly, in the present study, we propose a cultured hepatocyte model (icHep, shown in Fig. 1a) to evaluate biliary excretion using permeation assays, including both hepatic membrane transport and metabolism. Since cell adhesion proteins such as claudins regulate physiological bile canalicular lumen formation[11], we hypothesised that some cell adhesion proteins coated on microporous filters could induce the formation of bile canaliculi on the filter surface side. We screened the cell adhesion proteins essential for hepatocytes to form the bile canalicular lumen, synthesised such proteins in vitro, and attached them to a permeable support to mimic the hepatocyte surface. With this, we expect the formation of an open canalicular lumen face to a permeable support surface, enabling the evaluation of hepatobiliary disposition using permeation assays. We showed that the permeability of several drugs correlated well with the in vivo biliary excretion clearance in humans,

demonstrating the successful establishment of a hepatocyte culture model for the efficient evaluation of the hepatobiliary disposition of drugs.

## Results

**Identification of human claudin molecules involved in bile canaliculus formation**. To identify the claudin molecules involved in bile canaliculus formation, we first evaluated the expression level of each claudin molecule in the human hepatocarcinoma HepG2 and normal human liver tissue. HepG2 cells expressed 22 claudins. Compared to human liver tissue, claudins were generally expressed at a lower level in HepG2 cells, except for claudin-4, -6, -9, -11, -16, -17, -19, and -27 (Fig. 1b). The intracellular location of these claudins was investigated after transfecting each claudin-expressing plasmid with a myc tag attached to the N-terminus. As a result, 15 claudin molecules were found to be localized in the canalicular membrane, as evidenced by their colocalization with MRP2, a selective transporter protein in the canalicular membrane (Supplementary Fig. 1). In contrast, claudin-22 and claudin-6, -10b, -16, -20, and -24 were primarily expressed in the sinusoidal membrane and cytoplasm, respectively. Next, we investigated the changes in the number of bile canaliculi via co-immunostaining with MRP2 and ZO-1 (as a marker of mature bile canaliculi) after the transfection of claudin-expressing plasmids into HepG2 cells. All claudin molecules investigated were efficiently transfected into HepG2 cells by the lentiviral method (Supplementary Fig. 2). The expression of claudin-1, -2, -3, and -9 increased the number of bile canaliculi formed in HepG2 cells to $150 \pm 14$, $158 \pm 15$, $145 \pm 21$, and $142 \pm 15\%$, respectively, compared to mock cells (Fig. 1c). To clarify the involvement of these four claudin molecules in bile canaliculus formation, we used HeLa cells (as nonhepatocytes that express specific claudins; HeLa/CLDNs) by infecting lentiviral particles incorporating these four claudins and EGFP genes. A co-culture of the constructed HeLa/CLDN cells with HepG2 cells resulted in the formation of MRP2-containing bile canaliculi between HepG2 cells and EGFP-labelled HeLa/CLDN cells (Supplementary Fig. 3). It is noteworthy that HeLa/CLDN cells alone did not form bile canaliculi between them. In addition, HepG2 cells were grown on the upper side of a highly porous filter membrane[12] with HeLa/ CLDNs cells on the lower side to examine the formation of bile canaliculi upon intercellular contact across the filter membrane. As shown in Fig. 1d, bile canaliculi formed between these cells (i.e., across the filter membrane). These results demonstrated that the four claudin proteins selected were sufficient to induce bile canaliculus formation with adjacent hepatocytes.

**Construction of a hepatocyte culture system with hemi-bile canaliculi using a claudin-coated permeable support**. We hypothesised that the first contact between hepatocytes and claudin triggers bile canaliculus formation. Therefore, we prepared permeable supports coated with claudin-1, -2, -3, and -9 proteins and cultured primary human hepatocytes on these supports to induce the formation of bile canaliculi opening towards the support (Fig. 1a). Claudin proteins were prepared using the soluble membrane fragment (S-MF) method[13] in an *Escherichia coli* cell-free synthesis system. Western blotting (WB) analysis detected the cell-free product of each claudin as the main band, as indicated by the respective molecular weights (Supplementary Fig. 4). The synthesised cell-free claudin products were uniformly distributed throughout the support (Fig. 2a). Primary human hepatocytes (Donor ID. Hu1663) were cultured on supports coated with claudin-1, -2, -3, and -9. The localization of the bile canalicular membrane marker MRP2 was observed as a

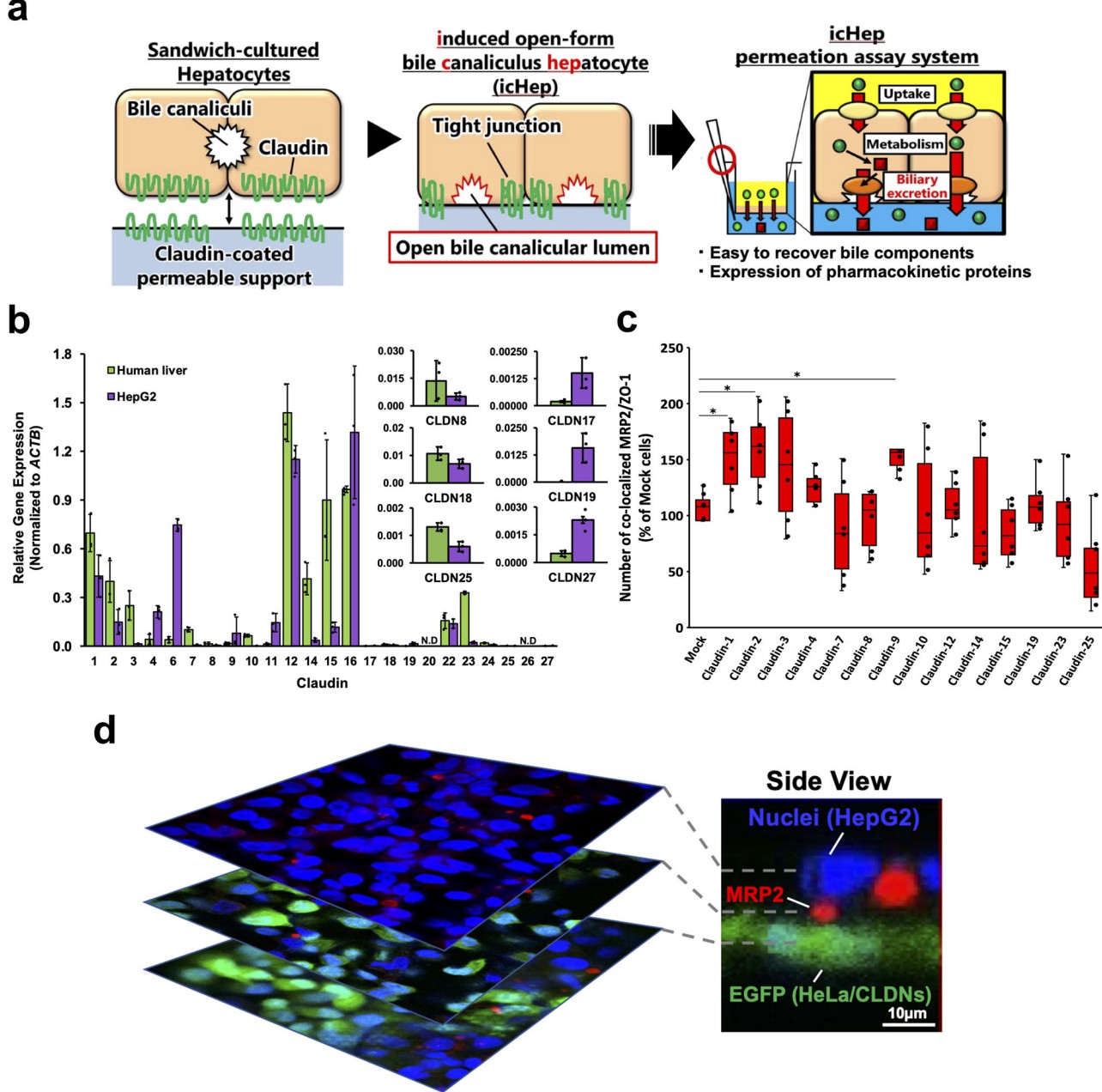

**Fig. 1 Identification of human claudin molecules involved in the formation of bile canaliculi. a** The outline of this study is shown. Claudin molecules sufficient to form bile canaliculi were identified and adhered to a permeable support to mimic the surface of hepatocytes. By culturing hepatocytes on this support, we established a hepatocyte culture system (icHep) that formed canalicular hemi-lumens between the hepatocytes and the permeable support surface. This system can be applied to the evaluation of hepatobiliary disposition using a transcellular permeation assay. **b** Gene expression of the claudin family in human liver tissue and HepG2 cells. The green and purple bars show the expression levels of claudins in the human liver and HepG2 cells, respectively. The bars represent the means of the corresponding group, and the error bars represent the S.D. ($n = 3$ biological replicate wells). Claudin gene expression was normalized to *ACTB*. **c** Effect of claudin on the formation of bile canaliculi in HepG2 cells. MRP2 and ZO-1 colocalizations were counted as mature bile canaliculi in both HepG2 cells overexpressing human claudins and mock cells. The box plots are represented as follows: the center line of the box indicates the median, the box indicates the upper and lower quartiles, and the whiskers show the maximum and minimum values ($n = 6$ biological replicate wells). Statistical significance was determined using Student's *t*-test; *$P < 0.05$. **d** Effect of co-culture with HeLa/CLDNs cells on the localization of the bile canaliculus marker MRP2 in HepG2 cells. Immunofluorescence staining of MRP2 (red) was observed upon co-staining for DRAQ5 (blue, nuclei), and EGFP (green, HeLa/CLDNs cells) in HepG2 cells co-cultured with HeLa/CLDNs cells. Scale bar = 10 μm.

semicircle opening to the contact surface between the hepatocytes and the claudin-coated support (Fig. 2b). In contrast, hepatocytes in contact with collagen-coated support formed closed bile canaliculi between adjacent hepatocytes. Accordingly, it was demonstrated that these four claudin proteins attached to the support membrane induce bile canaliculi in hepatocytes; hence,

we named this hepatocyte culture system induced open-form bile canaliculus hepatocyte (icHep). The icHep showed a hemi-lumen formation rate of $28.8 \pm 6.3\%$ (Fig. 2c). When the icHep was constructed with PXB-cells (donor: JFC), which are fresh hepatocytes derived from human liver chimeric mice[14], on supports individually coated with each claudin, hemi-lumen formation

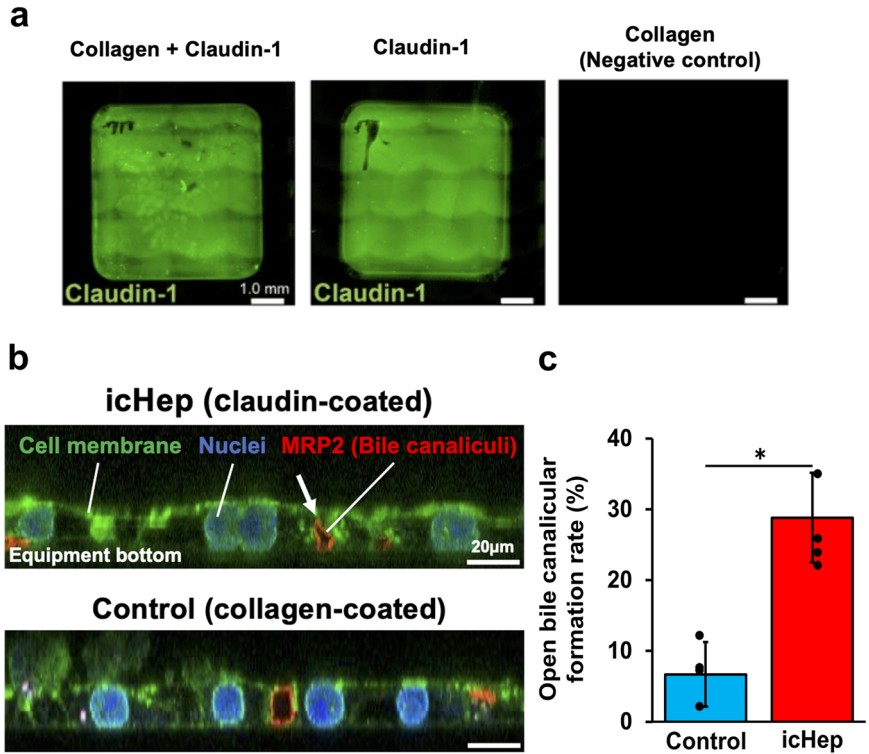

**Fig. 2 Claudin regulates the formation of bile canaliculi. a** Claudin-1 coating on the culture equipment. Immunofluorescence staining was observed for human claudin-1 (green) in each coated well; (left) collagen + claudin-1, (center) claudin-1, (right) collagen. Scale bar = 1.0 mm. **b** Manipulation of bile canalicular localization in primary human hepatocytes cultured on claudin-coated plates. Hepatocytes were cultured on a plate coated with (upper) human claudin-1, -2, -3, and -9 and (lower) collagen alone as a control. Immunofluorescence staining of MRP2 (red) was observed upon co-staining of the cell membrane marker WGA (green) and the nuclear marker DRAQ5 (blue). The white arrow indicates the bile canaliculi opened on the side of the culture equipment. Scale bar = 10 μm. **c** Number of localized bile canaliculi lumen face to culture plate (%) in primary human hepatocytes cultured on control (green) and claudin-coated (red) plates, respectively ($n = 4$ biological replicate wells). Data are represented as the means ± S.D. Statistical significance was determined using Student's $t$-test; *$P < 0.05$.

rates were 44.5 ± 4.8, 48.9 ± 12, 38.6 ± 8.1, and 35.9 ± 6.4% for claudin-1, claudin-2, claudin-3, and claudin-9, respectively (Supplementary Fig. 5). When all these four claudins were coated in a mixture, hemi-lumen formation rates were 49.2 ± 4.7% (donor: JFC) and 35.8 ± 6.3% (donor: HUM181001B), respectively. (Supplementary Fig. 5). Claudins juxtapose within the plasma membrane with claudins of the same (*cis*-interaction) or different subtypes on the adjacent plasma membrane (*trans*-interaction) to form tight junctions[15,16]. Thus, the claudin-coated surface serves as a paired pseudo-membrane for the formation of tight junctions in hepatocytes, facilitating the formation of hemi-lumens.

**Pharmacokinetic characterization of icHep**. A continuous monolayer of cells with a tight cell-to-cell contact is required for drug permeation assays using cell monolayers. Therefore, we evaluated the morphology of primary human hepatocytes and PXB-cells using a Transwell™ as a permeable support. Primary human hepatocytes we used are difficult to apply for permeability measurements because they form cell-free cavities. In contrast, when icHep was cultured with PXB-cells for 14 days, cavities were observed in the cell monolayer on day 1, polygonal structures characteristic of hepatocytes were observed on day 3, and closely packed hepatocyte monolayers were maintained until day 14 (Fig. 3a). Accordingly, we selected PXB-cells to construct a permeation assay system and evaluated the usefulness of icHep in evaluating the pharmacokinetic properties of drugs by functional comparison with conventional sandwich-cultured PXB-cells. First, albumin secretion and urea synthesis were measured to assess liver-specific functions of icHep. Albumin secretion and urea synthesis in icHep showed a linear increase throughout the 14-day culture period, and the amounts were significantly higher than those in the sandwich culture system (Fig. 3b, c). Next, gene expression changes of typical hepatic transporters and drug-metabolizing enzymes during the culture period were evaluated. Most of the investigated gene expression was relatively maintained during the culture period, while gene expression levels of some apical (BSEP), and basal (OATP1B1, OATP1B3) transporters and drug-metabolizing enzymes (CYP1A2, CYP2E1) declined by day 3 of culture. Each gene expression profile was comparable between icHep and conventional sandwich culture system (Fig. 3d, e). Furthermore, the cellular localization of apical (BSEP, MRP2, P-gp) and basolateral (OATP1B1, OATP1B3, OCT1, NTCP) transporters in icHep was also examined (Fig. 3f). All apical transporters were localized on the lateral membrane of the hemi-lumen, which was open to the membrane of Transwell™ permeable support. Basolateral transporters were mainly localized to the plasma membrane, except NTCP, OATP1B1, and OCT1, which were also observed in the cytoplasm. Furthermore, the activity of drug-metabolizing enzymes in icHep and sandwich-cultured PXB-cells obtained through the cocktail substrate method (see Supplementary Table 1) were measured for major CYPs (CYP1A2, CYP2B6, CYP2C9, CYP2C19, CYP2D6, CYP2E1, CYP3A4), and UGT1A1. icHep and sandwich-cultured PXB-cells derived from the same donor exhibited similar metabolic activities, consistent with their gene expression profiles (Fig. 3g). Moreover, donor-to-donor differences (JFC and HUM181001B) were identified in the activities of several drug-

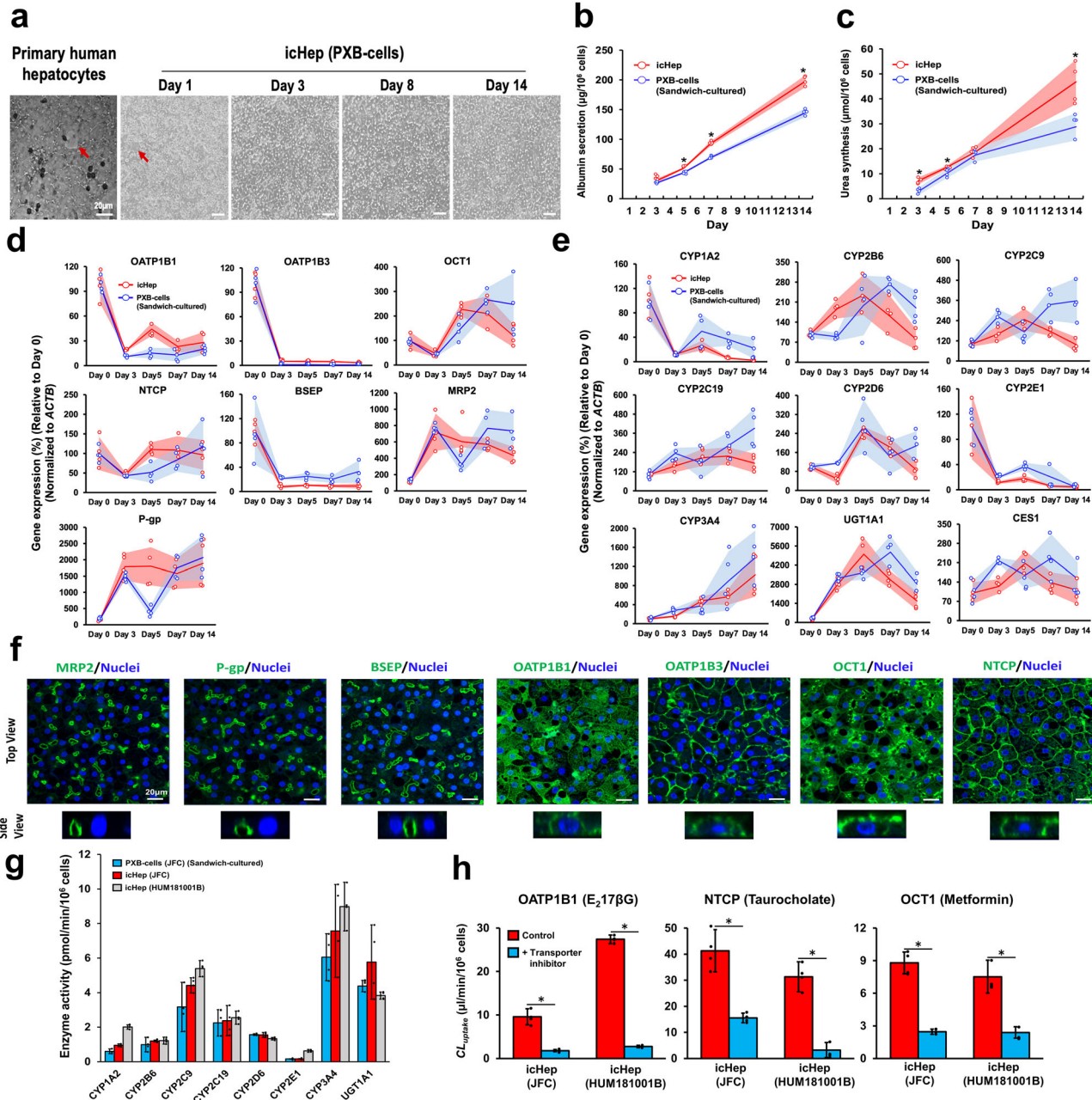

**Fig. 3 Pharmacokinetic functional evaluation of icHep.** PXB-cells were cultured on the permeable support coated with collagen and claudin-1, -2, -3, and -9. **a** Morphological change of icHep over time on the permeable support. Monolayer of primary human hepatocytes for 3 days culture and icHep for 1-, 3-, 8-, and 14-days culture were observed by bright field microscope. The red arrows show the cavities of the hepatocyte monolayer. Bar = 20 μm. **b**, **c** (**b**) Albumin secretion and (**c**) urea synthesis in icHep (red) and PXB-cells (blue) sandwich cultured on the collagen-coated permeable support were measured on day 3, 5, 7 and 14. The dot plots represent the means of the groups and the shaded error bars represent the S.D. ($n$ = 4 biological replicate wells). **d**, **e** The gene expression of (**d**) drug transporters and (**e**) drug-metabolizing enzymes in icHep (red) and PXB-cells (blue) sandwich cultured on the collagen-coated permeable support cultured for 0, 3, 5, 7, and 14 days was measured via quantitative PCR. The dot plots represent the means of the groups and the shaded error bars represent the S.D. ($n$ = 4 biological replicate wells). Gene expressions were normalized to *ACTB* and were shown relative to day 0. **f** Localization of hepatic drug transporters in icHep. Immunofluorescence staining was performed for canalicular [BSEP, MRP2, and P-gp: green)] and sinusoidal transporters (NTCP, OATP1B1, OATP1B3, and OCT1; green), and the nuclear marker DRAQ5 (blue). Scale bar = 20 μm. **g** Drug metabolic activity in icHep. Drug metabolic activity on day 7 in icHep (JFC: red bars, HUM181001B: green bars) and PXB-cells (JFC: blue bars) sandwich cultured on the collagen-coated permeable support. Bars show the amount of metabolite produced after exposure of cells to the substrate cocktail of each drug-metabolizing enzyme ($n$ = 4 biological replicate wells). **h** Uptake activity of drugs in icHep. Drug uptake activity in icHep was investigated on day 7 using PXB-cells derived from two human liver donors (JFC and HUM181001B). Bars represent the uptake of typical substrates (1 μCi/ml) of the uptake transporters (OATP1B1: E$_2$17βG, OCT1: metformin, NTCP: taurocholate) in the absence (red bars) or presence (green bars) of their respective transporter inhibitors [OATP1B1: rifampicin (10 μM), OCT1: quinidine (100 μM), NTCP: cyclosporine A (10 μM)]. ($n$ = 4 biological replicate wells). Data are presented as the means ± S.D. Statistical significance was determined using Student's $t$-test; *$P$ < 0.05.

metabolizing enzymes (CYP1A2, CYP2C9, CYP2E1, CYP3A4, and UGT1A1) in icHep (Fig. 3g).

Next, we evaluated the ability of basolateral transporters (OATP1B1, OCT1, NTCP) in icHep to uptake typical substrates (OATP1B1: $E_2 17\beta G$, OCT1: metformin, NTCP: taurocholate). Both icHep from the two investigated human liver donors (JFC and HUM181001B) had the ability to uptake each transporter typical substrate. In particular, a 2.85-fold difference was observed in OATP1B1 substrate uptake clearance between two donors (JFC: $27.4 \pm 1.0$, HUM181001B: $9.60 \pm 1.8\ \mu l/min/10^6$ cells). In addition, the observed increase in substrate uptake was counteracted upon treatment with inhibitors of each transporter, suggesting that icHep maintained functional hepatic uptake transporter activity (Fig. 3h). These results demonstrate that icHep possess a human-type liver function necessary to assess the hepatobiliary disposition of drugs.

**Evaluation of biliary excretion of drugs using permeation assays with icHep.** Since icHep form an open hemi-lumen at the interface between hepatocytes and a permeable support with immobilised claudins, it is expected to be applicable to biliary excretion studies of drugs using permeation assays. We first measured the degree of paracellular leakage of icHep and PXB-cells (as a control group) monolayers cultured on Transwell™ membranes as a permeable support. Minimal permeation of a fluorescent probe TD4 (4 kDa, fluorescent dextran) from the donor to receiver chambers was observed in both cell monolayers on days 3, 5, and 8 of culture (Fig. 4a). The permeability of the probe decreased significantly in the icHep monolayer compared to the control during the culture period of up to 8 days. Although the transepithelial electrical resistance (TEER), a measure of ion permeability, was comparable in both cell monolayers (Fig. 4b), a stronger intercellular barrier for macromolecules was formed in the icHep cell layer. The MRP2-mediated biliary excretion activity of icHep was also evaluated using permeation assays. Here, CDFDA (5,6-carboxy-2',7'-dichlorofluorescein diacetate) was selected, because it is taken up by hepatocytes via passive diffusion, followed by metabolism by intracellular esterase to the fluorescent compound CDF, which is excreted into the bile by MRP2[17]. When CDFDA was loaded onto the sinusoidal side (upper chamber), the appearance of CDF on the bile side (lower chamber) of icHep increased to $147 \pm 15\%$ of the control group. Furthermore, treatment with the MRP2 inhibitor benzbromarone reduced this increase to the same level as the control group (Fig. 4c). Therefore, the biliary excretion of drugs and hepatic metabolism can be evaluated using permeation assays with icHep. In addition, permeability assays were performed for representative substrates of biliary excretion transporters (BSEP: taurocholate, MRP2: $E_2 17\beta G$, and P-gp: digoxin). The permeability of taurocholate, $E_2 17\beta G$, and digoxin to the bile side was significantly increased in icHep from two donors (JFC and HUM181001B) compared to the control (Fig. 4d, Supplementary Fig. 6). Moreover, the increase in permeation was suppressed upon treatment with inhibitors of each transporter. The results of our permeation and transporter inhibition studies demonstrated that a high-throughput evaluation of the hepatobiliary disposition of drugs is possible using icHep.

**Prediction of human in vivo drug biliary excretion using permeation assays with icHep.** We studied the predictability of human in vivo biliary excretion using drug permeation assays with icHep from two donors and compared them with conventional sandwich-cultured PXB-cells. Seven drugs whose human in vivo biliary excretion clearances are known (cimetidine, digoxin, erythromycin, methotrexate, nafcillin, SN-38, and

vincristine) were evaluated. In the permeation assays using icHep, the permeability coefficient of each drug was significantly increased compared to that of the control and decreased in the presence of their respective transporter inhibitors (Figs. 4d, 5a, b, Supplementary Fig. 6). Subsequently, the in vivo biliary excretion clearance, plasma protein-unbound fraction, and blood/plasma ratio of each drug were obtained from the literature, and their in vivo biliary excretion intrinsic clearances in human were estimated using Eq. (4) (see Methods section). The pharmacokinetic and biliary excretion parameters of each drug are summarised in Table 1. The predicted human in vivo biliary excretion intrinsic clearance of the drugs in icHep exhibited a good correlation with the corresponding human in vivo values [icHep (JFC): $r = 0.969$ and $p = 0.001$, icHep (HUM181001B): $r = 0.966$ and $p = 0.001$] (Fig. 5d, e). Furthermore, the BEI of each drug was calculated as an index of the drug biliary excretion activity using sandwich-cultured PXB-cells for comparison with icHep. However, the calculated BEIs were variable depending on the incubation time of each drug to PXB-cells, and did not show a constant value (Fig. 5c). In addition, the prediction accuracy of in vivo clearance estimated from BEI in PXB-cells differed depending on the incubation time of each drug (2 min incubation: $r = 0.498$ and $p = 0.255$, 5 min incubation: $r = 0.994$ and $p = 0.001$, 10 min incubation: $r = 0.952$ and $p = 0.001$, 20 min incubation: $r = 0.952$ and $p = 0.001$) (Supplementary Fig. 7). Thus, our icHep permeation assay system efficiently enables more reliable prediction of the human in vivo biliary excretion of drugs.

**Evaluation of drug-induced cholestasis (DIC) involving metabolic processes using icHep.** Inhibition of BSEP by drugs and/or their metabolites can lead to DIC with intracellular accumulation of bile acids and subsequent cholestasis, leading to severe liver injury. Since DIC is often associated with drug metabolism processes, it is desirable to evaluate using SCHs that maintain liver physiology. We have previously reported using SCHs that BSEP is inhibited by candesartan cilexetil (CIL), but not by its active metabolite candesartan (CAN)[18]. Here, we investigated the applicability of icHep as an evaluation system for DIC using CIL. Treatment of the cells with CIL alone did not affect biliary excretion clearance of BSEP substrate taurocholate (Fig. 6a). This is due to the rapid hydrolysis of CIL to CAN by esterase in hepatocytes. On the other hand, treatment of the cells with CIL in the presence of esterase inhibitor diisopropyl fluorophosphate (DFP) resulted in the decreased biliary excretion clearance of taurocholate, accompanied by increased intracellular accumulation of taurocholate (Fig. 6a, b). Furthermore, intracellular CIL and CAN were increased and decreased, respectively, by DFP treatment in a concentration dependent manner (Fig. 6c). A significant correlation was observed between the decrease in biliary excretion clearance of taurocholate and the increase in intracellular CIL concentration ($r = -0.984$, $p = 0.01$), whereas there was no correlation with intracellular CAN concentration ($r = 0.274$, $p = 0.66$) (Fig. 6d, e). These results indicate that the icHep permeation assay system can be useful to evaluate DIC, including intrahepatic drug metabolism.

## Discussion

Evaluating the biliary excretion of drug candidates at the preclinical stage is a crucial problem to address during drug discovery and development. In the present study, we demonstrated that human hepatocytes formed an open-form bile canalicular lumen at the surface of permeable support membranes coated with specific claudin proteins (icHep). The established icHep system maintained the expression and localization of drug-metabolizing enzymes and transporters. Moreover, the biliary

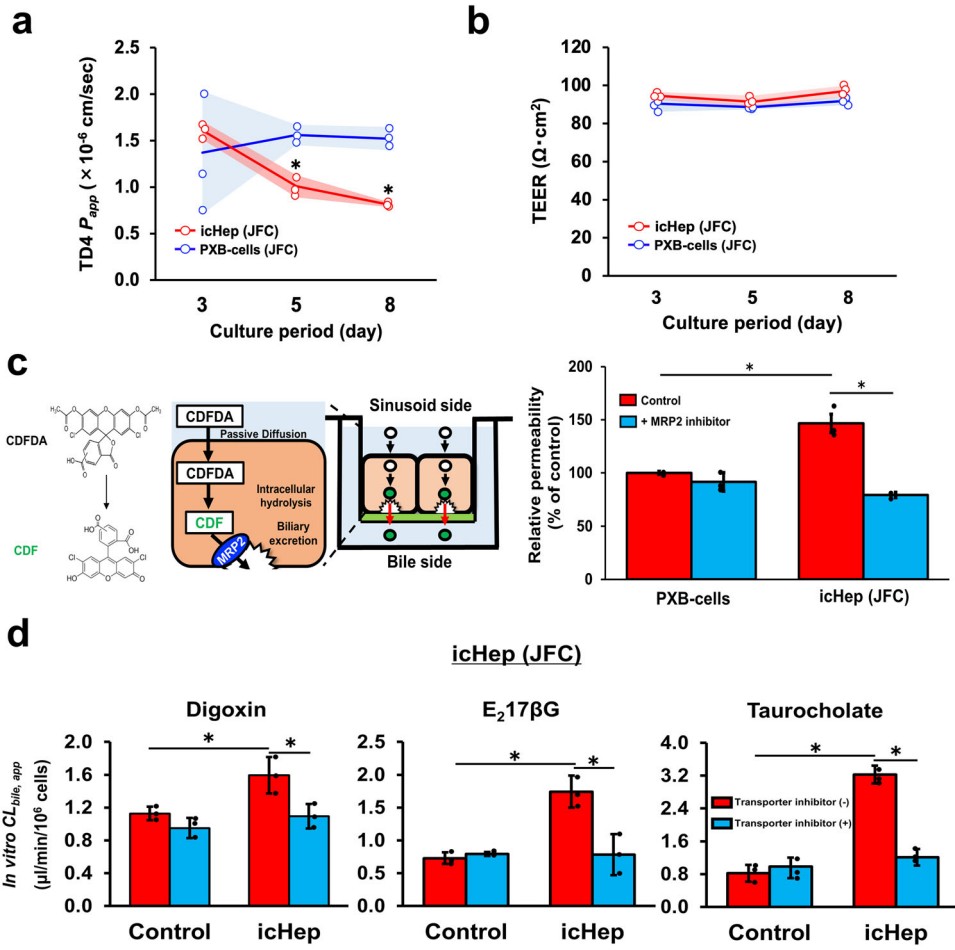

**Fig. 4 Evaluation of drug biliary excretion using permeation assays with icHep. a, b**, Formation of tight junctions in icHep and PXB-cells. (**a**) The $P_{app}$ values of the paracellular marker TD4 and (**b**) the TEER (Trans-epithelial electrical resistance) were measured in icHep (red) and PXB-cells (blue) cultured on the permeable support for 3, 5, and 8 days. The dot plots represent the means of the groups and the shaded error bars represent the S.D. ($n = 3$ biological replicate wells). Statistical significance compared with control PXB-cells cultured on the collagen-coated permeable support at the same time points was determined using Student's $t$-test; $*P < 0.05$. **c** MRP2-mediated biliary transport of CDF in icHep. icHep were cultured for 7 days on the permeable support and the biliary export of CDF across the cells was measured from the sinusoidal side to the bile side. (Upper panel) Schematic diagram of the permeation assay of the MRP2 substrate CDF using icHep. (Lower panel) icHep were incubated with CDFDA (5 µM) on the sinusoidal side for 120 min in the absence (red) or presence (green) of the MRP2 inhibitor benzbromarone (100 µM). **d** Biliary transporter-mediated transport of each substrate in icHep (JFC). Briefly, icHep were cultured on a permeable support for 7 days. The biliary transport of digoxin, $E_2$17βG, and taurocholate across the cells was measured from the sinusoidal side to the bile side. icHep were incubated with (left) [$^3$H]digoxin (1 µCi/ml), (center) [$^3$H]$E_2$17βG (1 µCi/ml), or (right) [$^3$H]taurocholate (1 µCi/ml) on the sinusoidal side for 120 min in the absence (red bars) or presence (green bars) of (left) the P-gp inhibitor zosuquidar (5 µM), (center) MRP2 inhibitor benzbromarone (100 µM), or (right) BSEP inhibitor chlorpromazine (30 µM), respectively. Data are presented as the means ± S.D. ($n = 3$ biological replicate wells). Statistical significance was determined using the Tukey-Kramer test; $*P < 0.05$.

excretion of drugs and their metabolites can be easily measured upon sampling from the receiver-side chamber in a time-dependent manner. Furthermore, we showed that transporter-mediated biliary excretion of drugs and bile acids could be evaluated using permeation assays with the icHep cell culture system, which showed the successful estimation of in vivo biliary excretion clearance from membrane permeability as calculated using the system.

SCHs are currently used to predict the biliary excretion of drugs as a representative in vitro method[19]. Although SCHs have the merit of maintaining the expression of drug-metabolizing enzymes and transporters for a long enough time to study drug disposition, they have several drawbacks in evaluating hepato-biliary drug disposition. For example, when the excreted drugs accumulate in the closed lumen, a rapid equilibrium results between the cells and the lumen. In fact, the calculated BEI and in vivo biliary excretion intrinsic clearance of the test drugs varied

with drug treatment time in sandwich-cultured PXB-cells (Fig. 5c, Supplementary Fig. 7). Moreover, the depletion of Ca$^{2+}$ in the culture medium affects cell viability and NTCP activity[20]. In contrast, icHep enables the continuous recovery of a drug and its metabolites as they become secreted into the bile side chamber; therefore, it can overcome the issues associated with SCHs and be applied to the robust screening of the biliary excretion of drug candidates. On the other hand, in vitro biliary intrinsic clearance ($CL_{bile, int}$) was calculated by subtracting the in vitro $CL_{bile, app}$ of control cells from that of icHep. This subtraction leads to increased variance of the calculated biliary excretion values. Improvement of formation of open canaliculus is desirable to increase the predictivity. In addition, PXB-cells used for constructing icHep contain about 10% mouse-derived cells[21]. Therefore, some of the obtained biliary transport includes mouse-derived reactions. Further study is required to establish icHep with pure human hepatocytes such as improved primary cultured

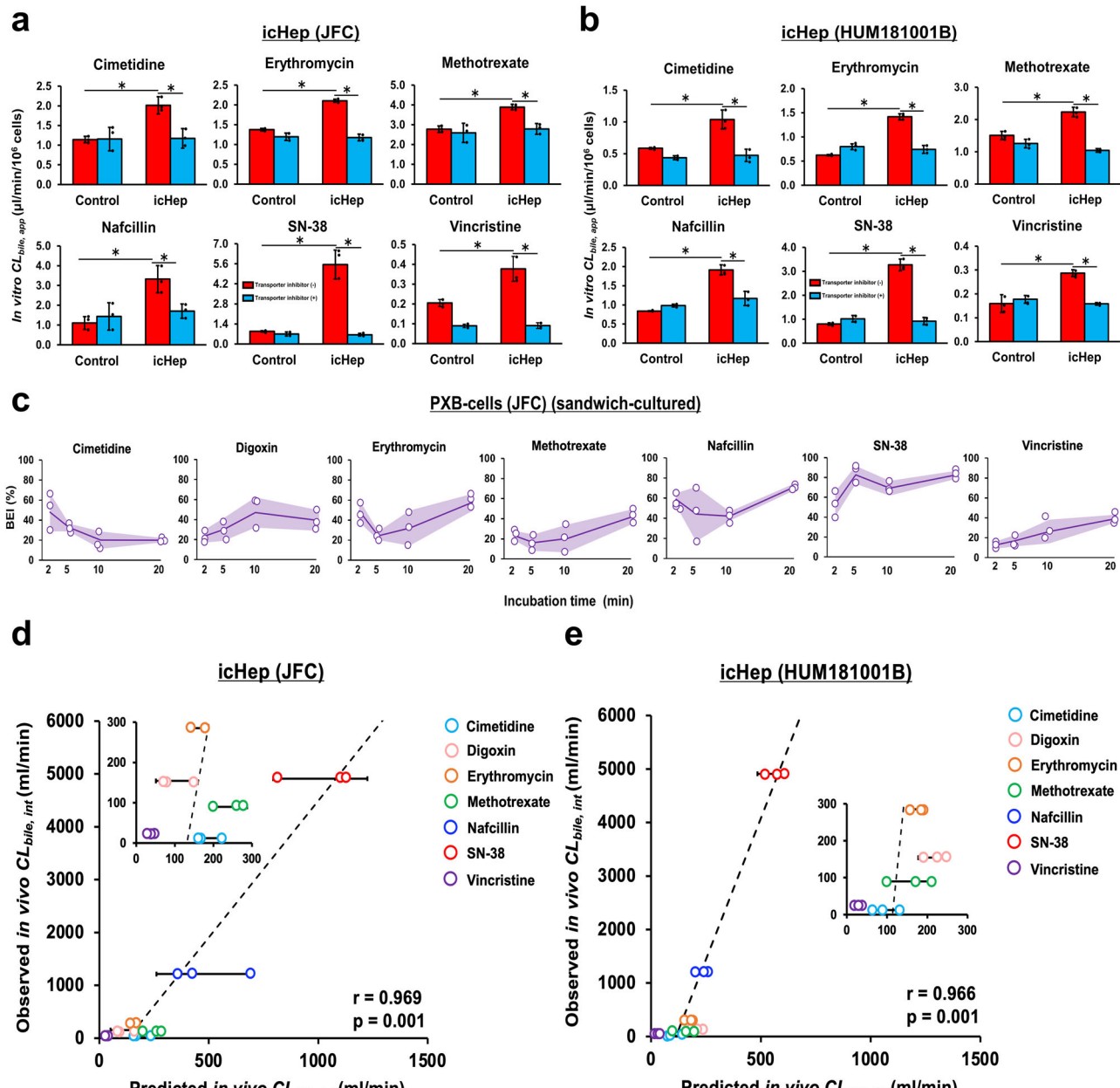

**Fig. 5 Comparison of biliary excretion clearance between icHep in vitro in those reported in humans in vivo. a, b** The biliary transport of MRP2 and P-gp substrates across the icHep [(**a**) JFC, (**b**) HUM181001B] was measured from the sinusoidal side to the bile side. icHep were incubated with 1 μM each of a mixture of substrates (cimetidine, erythromycin, methotrexate, nafcillin, SN-38, and vincristine) on the sinusoidal side for 120 min in the absence (red bars) or presence (green bars) of a mixture of the MRP2 inhibitor benzbromarone (100 μM) or the P-gp inhibitor zosuquidar (5 μM). **c** Time-dependent BEI variation of substrates with known human in vivo biliary clearance in sandwich-cultured PXB-cells. PXB-cells (JFC) were incubated with 1 μM each of a mixture of substrates (cimetidine, digoxin, erythromycin, methotrexate, nafcillin, SN-38, and vincristine) for 2, 5, 10, and 20 min. The BEI for each substrate was calculated using Eq. (8). The dot plots represent the means of the groups and the shaded error bars represent the S.D. ($n = 3$ biological replicate wells). **d, e** The correlation between in vitro [(**d**) icHep (JFC) and (**e**) icHep (HUM181001B)] and in vivo biliary clearance of test substrates was evaluated. Data are presented as the means ± S.D. ($n = 3$ biological replicate wells). Statistical significance was determined using the Tukey-Kramer test and Pearson's correlation analysis; *$P < 0.05$.

hepatocytes[22,23] and iPS-derived hepatocytes[24]. On the other hand, claudin was narrowed down using HepG2 cells (Fig. 1c). Since claudin genes expressed in HepG2 cells have different patterns from human hepatocytes (Fig. 1b), it is possible to construct icHep with higher formation of open-form canaliculus by further optimizing the types and combinations of claudin proteins using human hepatocytes.

One major clinical regulatory issue to be overcome is drug interactions on hepatic drug-metabolizing enzymes and

transporters. icHep confirmed to express enzymes and transporters important for evaluating the risk associated with drug interactions and drug-induced cholestasis. Currently, the inhibitory potential of parental drugs on bile canalicular membrane transporters, such as MRP2 and BSEP, is being evaluated cell-free system using in vitro transporter-expressing membrane vesicles. However, in addition to the parent drug, its metabolites formed in hepatocytes may contribute to the inhibition of these transporters. Metabolite identification and the measurement of their

**Table 1 Pharmacokinetic and biliary excretion parameters of drugs used in this study.**

| Substrate | Biliary excretion clearance (ml/min) | | | | | Biliary excretion clearance (μl/min/10⁶ cells) | | | | | Pharmacokinetic parameters | |
|---|---|---|---|---|---|---|---|---|---|---|---|---|
| | In vivo | | | | | In vitro | | | | | | |
| | Predicted | | | Observed | | Observed | | | | | | |
| | icHep (JFC) $CL_{bile,int}$ [a] | icHep (HUM181001B) | PXB-cells (JFC) | $CL_{bile,int}$ [b] | $CL_{bile}$ [c] | icHep (JFC) $CL_{bile,int}$ [d] | $CL_{bile,app}$ [e] | icHep (HUM181001B) $CL_{bile,int}$ [d] | $CL_{bile,app}$ [e] | PXB-cells (JFC) BEI (%) | $f_p$ [f] | $R_b$ [g] |
| Cimetidine | 188 ± 32.8 | 97.8 ± 35.1 | 60.3 ± 8.17 | 12.3 | 9.9[31] | 0.869 ± 0.0152 | 2.01 ± 0.217 | 0.453 ± 0.163 | 1.04 ± 0.149 | 19.6 | 0.81[32] | 0.97[32] |
| Digoxin | 101 ± 50.0 | 211 ± 33.7 | 152 ± 61.5 | 154 | 106[33] | 0.467 ± 0.232 | 1.59 ± 0.222 | 0.979 ± 0.156 | 2.11 ± 0.146 | 37.1 | 0.76[34] | 0.8[35] |
| Erythromycin | 158 ± 15.6 | 171 ± 15.6 | 310 ± 98.9 | 285 | 80.3[36] | 0.732 ± 0.0721 | 2.11 ± 0.0406 | 0.793 ± 0.0723 | 1.42 ± 0.0635 | 54.8 | 0.305[37] | 0.75[38] |
| Methotrexate | 241 ± 46.9 | 157 ± 57.9 | 172 ± 50.0 | 90.0 | 39.9[39] | 1.11 ± 0.217 | 3.89 ± 0.138 | 0.73 ± 0.268 | 2.23 ± 0.148 | 40.6 | 0.46[40] | 0.79[41] |
| Nafcillin | 479 ± 218 | 232 ± 30.1 | 608 ± 101 | 1213 | 125[42] | 2.22 ± 1.01 | 3.32 ± 0.689 | 1.07 ± 0.14 | 1.91 ± 0.132 | 70.9 | 0.123[43] | 0.55[43] |
| SN-38 | 1009 ± 215 | 532 ± 47.4 | 1099 ± 280 | 4910 | 42.5[44] | 4.67 ± 0.996 | 5.55 ± 1.01 | 2.46 ± 0.219 | 3.27 ± 0.24 | 81.2 | 0.00087[45] | 1.25[45] |
| Vincristine | 37.3 ± 11.2 | 27.6 ± 9.46 | 168 ± 29.4 | 25.5 | 12.9[46] | 0.173 ± 0.0520 | 0.377 ± 0.0634 | 0.128 ± 0.0438 | 0.287 ± 0.039 | 40.4 | 0.51[47] | 1.2[47] |

[a]Predicted human in vivo biliary excretion intrinsic clearance estimated from the icHep permeation assay and biliary excretion assay using sandwich-cultured PXB-cells.
[b]Human in vivo biliary excretion intrinsic clearance estimated from the reported values.
[c]Reported human in vivo biliary excretion clearance.
[d]Human in vitro biliary excretion intrinsic clearance calculated using the icHep permeation assay.
[e]Apparent human in vitro biliary excretion clearance calculated using the icHep permeation assay.
[f]Reported value of the unbound fraction of the plasma protein of the compound.
[g]Reported value of the blood/plasma concentration ratio of compound. Data are represented as means ± S.D. (n = 3 biological replicate wells).

intracellular concentrations are needed to evaluate their inhibitory effect on biliary transporters; however, their precise estimation is challenging. On the other hand, icHep addresses this problem because it enables the evaluation of drug-induced changes in biliary excretion by simply applying drugs and bile acid in the donor side chamber and measuring appeared bile acid in the receive-side chamber. Therefore, we examined the applicability of icHep to the evaluation of biliary excretion inhibition which is affected by intracellular drug metabolism process, using CIL, whose BSEP inhibitory efficacy is attenuated by intracellular metabolism to CAN by esterase. Co-exposure of DFP and CIL decreased biliary clearance of taurocholate with increasing intracellular concentration of CIL. This observation is explained by the inhibition of BSEP-mediated efflux of taurocholate due to decreased intracellular CAN by DFP. Therefore, icHep is applicable for assessment of DIC, including intrahepatic drug metabolism using permeation assays.

In drug development studies, the human safety of drug candidates cannot be sufficiently predicted using preclinical safety studies with experimental animals. One reason for this is that the metabolites produced are specific to humans and that higher levels of metabolites are found in humans than in experimental animals[25]. In the Safety Testing of Drug Metabolites guidance released by the US-Food and Drug Administration (FDA), the safety of drug metabolites also needs to be evaluated before large-scale clinical trials[26]. To evaluate the pharmacokinetics of drug metabolites produced in the liver, there is demand for an in vitro method that can be used to predict how much of the metabolites formed in the liver are back fluxed into the blood, which may cause adverse events. icHep provides easy access to secreted metabolites into the blood and bile side chambers. Therefore, icHep may be used to predict blood levels of these metabolites. In addition, biliary-excreted parental drugs and metabolites are reabsorbed in the gastrointestinal tract, in a process called enterohepatic circulation. It has a significant impact on drug pharmacokinetics, efficacy, and safety. MPS is useful for studying inter-organ relationships; however, enterohepatic circulation is difficult to establish using currently available cell culture methods. It is expected that MPS, including organ-on-a-chip systems equipped with icHep and intestinal cells and intestinal microbiome will enable to evaluate complicated enterohepatic circulation.

In conclusion, we have shown that drug biliary excretion clearance can be estimated using icHep, which form a semicircle open to the contact surface between hepatocytes and a claudin-coated permeable support membranes, enabling the continuous collection of biliary-secreted drugs. We believe this approach will be useful for evaluating the pharmacokinetics and safety of drugs during the preclinical stages of drug discovery and development.

## Methods

**Materials.** Cryopreserved hepatocyte recovery medium (CHRM), primary hepatocyte maintenance supplement (CM3000), and William's E medium (WEM) were purchased from Thermo Fisher Scientific (Waltham, MA, USA). Human 3 donor pooled liver total RNA (Lot ID. 1402003) was obtained from Clontech (Mountain View, CA, USA). Insulin-transferrin-selenium (ITS) liquid medium was obtained from Sigma-Aldrich (St. Louis, MI, USA). Matrigel and Transwell™ permeable supports made of polyethylene terephthalate (pore size 0.4 μm) were obtained from Corning (Cambridge, MA, USA). [³H]E₂17βG (50 Ci/mmol), [³H]taurocholate (15.4 Ci/mmol), [³H]digoxin (23.8 Ci/mmol), and [¹⁴C]metformin (58 mCi/mmol) were purchased from PerkinElmer (Waltham, MA). All other chemicals used were commercially available and of reagent grade.

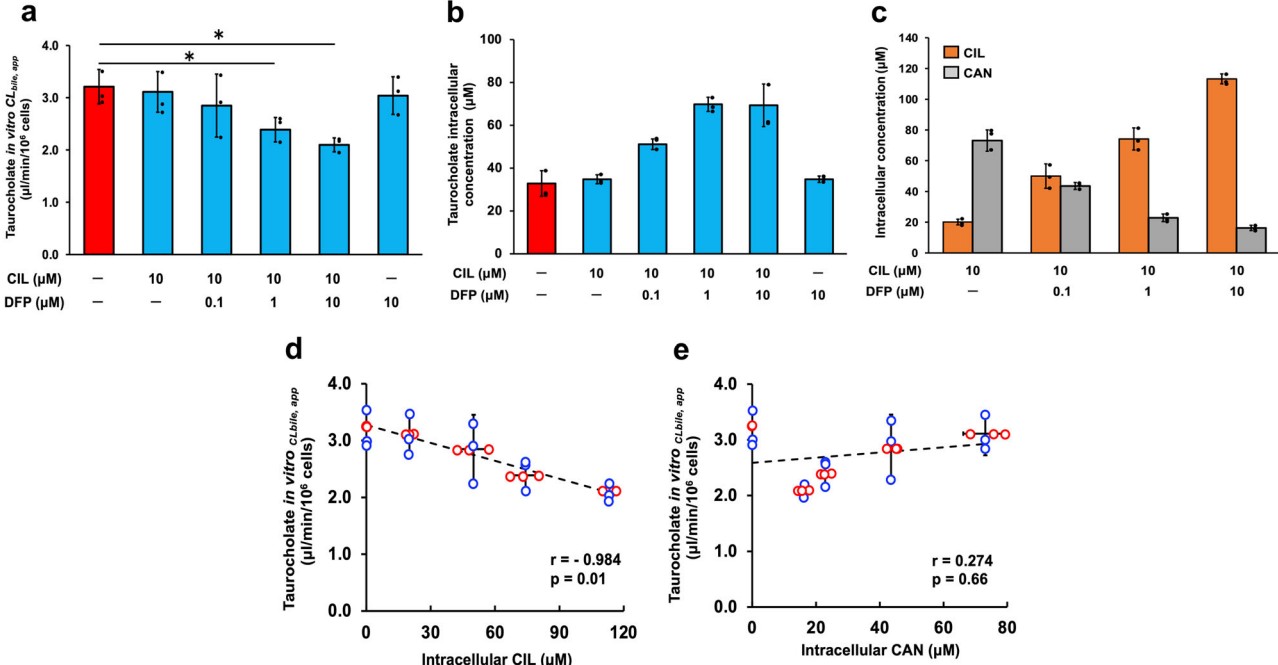

**Fig. 6 Drug-bile acid interactions involving drug metabolism and BSEP using icHep. a** The biliary transport of taurocholate across the icHep was measured from the sinusoidal side to the bile side. icHep were incubated with 1 μM taurocholate on the sinusoidal side for 120 min in the absence (red bar) or presence (green bars) of CIL (10 μM) and/or DFP (0.1, 1, and 10 μM). **b** Intracellular accumulation of taurocholate after 120 min of substrate permeation in the absence (red bar) or presence (green bars) of CIL (10 μM) and/or DFP (0.1, 1, and 10 μM). **c** Intracellular accumulation of CIL (orange bars), and CAN (silver bars) after 120 min of substrate permeation in the absence or presence of DFP (0.1, 1, and 10 μM). **d, e** The correlation between the biliary transport of taurocholate across the icHep and intracellular accumulation of (**d**) CIL or (**e**) CAN was evaluated. Data are presented as the means ± S.D. ($n = 3$ biological replicate wells). Statistical significance was determined using the Tukey-Kramer test and Pearson's correlation analysis; *$P < 0.05$.

**Cell culture**. HepG2, HEK293T, and HeLa cells were obtained from the American Type Culture Collection (ATCC; Manassas, VA, USA). HepG2 cells were cultured in DMEM containing 10% FBS, 1% penicillin-streptomycin, and 1% non-essential amino acids (NEAA). HeLa cells were cultured in DMEM containing 10% FBS and 1% penicillin-streptomycin. HEK293T cells were cultured in DMEM containing 10% FBS, 1% penicillin-strepto-mycin, and 110 mg/l sodium pyruvate. All the cell lines were cultured according to the instructions provided by the ATCC and negative for mycoplasma contamination.

**Preparation of HepG2 cells expressing human claudins fused with myc tag**. To construct expression vectors of human claudins (-1, -2, -3, -4, -6, -7, -8, -9, -10a, -10b, -11, -12, -14, -15, -16, -19, -20, -22, -23, -24, and -25) fused with the myc tag sequence (EQKLISEEDL) at the N-terminus, DNA fragments of myc tag fusion claudin were inserted into a pcDNA3.1 (+) vector (Thermo Fisher Scientific). The constructed plasmids were transfected into HepG2 cells using Lipofectamine 3000 (Invitrogen), according to the manufacturer's instructions.

**Bile canaliculus counting assay**. To evaluate the formation of bile canaliculi, the number of mature bile canaliculi was determined by counting the number of co-stains of MRP2 expressed on the canalicular membrane and the scaffold protein ZO-1. For immunostaining, cells were fixed in 4% paraformaldehyde for 15 min, permeabilised with 0.1% Triton X-100 for 5 min, and blocked in 2% bovine serum albumin (BSA) for 1 h at 18–22 °C. After incubation in the blocking solution, the cells were incubated with primary antibodies at 4 °C overnight. Primary antibodies were detected using secondary antibodies at room temperature for 1 h. Hoechst 33342 (Thermo Fisher Scientific) was used to

counterstain nuclei for 10 min at room temperature. The antibodies used for immunostaining are summarised in Supplementary Table 3.

**Production and titres of lentiviral particles**. The FG12 lentiviral vector (Addgene, Watertown, MA, USA) was used for gene transfection of human claudin-1, -2, -3, and -9 (hCLDNs) in HeLa and HepG2 cells. Each DNA fragment of human claudin was inserted into the FG12 vector. Twenty-four hours before gene transfection, HEK293T cells were seeded on a 6-well plate at a density of $0.95 \times 10^5$ cells/cm². The transfection mixture of FG12/hCLDNs, the expression plasmid of the vesicular stomatitis virus (VSV-G), and the packaging plasmid psPAX2 were then added to the cells using Lipofectamine 3000 reagent. The spent culture medium was collected 72 h after transfection, centrifuged at $150 \times g$ at 4 °C for 5 min, and the supernatant was stored at −80 °C. Lentivirus titres were measured using a qPCR Lentivirus Titer Kit (Applied Biological Materials, Vancouver, Canada) according to the manufacturer's instructions.

**Co-culture of claudin-expressing HeLa (HeLa/CLDNs) and HepG2 (HepG2/CLDNs) cells**. Expression plasmids encoding hCLDNs were transduced into HeLa and HepG2 cells by culturing for 24 h in DMEM containing a 25% FG12/hCLDNs lentivirus solution (titres: claudin-1, $6.36 \times 10^4$ IU/ml; claudin-2, $1.38 \times 10^5$ IU/ml; claudin-3, $1.86 \times 10^6$ IU/ml; and claudin-9, $9.24 \times 10^4$ IU/ml). Lentivirus-infected cells were cultured in a glass plate or rear-back culture through a 3.0-μm pore-size honeycomb film (HCF)[12]. In the case of mixed cultures, virus-infected HeLa and HepG2 cells were separated using a 0.1% trypsin/EDTA solution, and each cell type was seeded at densities of $12.5 \times 10^4$ and $4.2 \times 10^4$ cells/cm² on the same plate, respectively. In front-to-back cultures, HeLa/CLDNs

cells ($0.25 \times 10^5$ cells/cm$^2$) were seeded on the back of an HCF. After 16 h, the film was turned over, HepG2/CLDNs cells ($4.0 \times 10^5$ cells/cm$^2$) were seeded, and cultured for 4 days.

**Cell-free protein synthesis and purification of human claudin-1, -2, -3, and -9 proteins**. In human claudin-1, -2, -3, and -9-expressing pcDNA3.1 (+) plasmids, the N-termini of the protein products were fused with a modified natural polyhistidine affinity tag (N11, MKDHLIHNHHKHEHAHAEH), and a TEV protease recognition sequence was inserted between polyhistidine affinity tag and claudin sequences. For the in vitro synthesis of human claudin-1, -2, -3, and -9 proteins, the soluble-membrane fragment (S-MF) method[13] was used using a cell-free protein synthesis system (Musaibo-Kun, Taiyo Nippon Sanso, Tokyo, Japan). In detail, mixed micelles were prepared from a mixture of 67 mg/ml lipids [5% (w/w) cholesterol and 95% (w/w) egg PC] and 100 mg/ml digitonin via sonication until the solution became clear. Human claudin-1, -2, -3, and -9 proteins were then produced using the S-MF method with 3.959% (w/v) PEG-8000 in the presence of 5% (v/v) 50 ng/μl template plasmids and 10% (v/v) the lipid/detergent mixed micelles at 30 °C for 5 h with shaking. Claudin proteins produced in the reaction solution were collected in the supernatant via centrifugation at $15,000 \times g$ at 4 °C for 1 min. Subsequently, the claudin proteins were adsorbed in a Ni Sepharose 6 Fast Flow affinity resin (Cytiva, Tokyo, Japan), washed with a wash buffer (50 mM Tris-HCl [pH 7.0] containing 0.05% βDDM, 0.002% CHS, 20 mM imidazole, and 400 mM NaCl), and then eluted with the same buffer containing 500 mM imidazole. The N11 tag of the claudin protein was cleaved upon digestion with His-tagged TEV protease at 4 °C overnight in a dialysis buffer (50 mM Tris-HCl [pH 7.0] containing 0.05% βDDM, 0.002% CHS, and 400 mM NaCl). The N11 tag and His-tagged TEV protease were removed using reverse immobilised metal ion affinity chromatography, and the His-tag-free claudin protein was collected in the flow-through fraction. The synthetic product of the claudin protein was evaluated via western blotting. Densitometry analysis of the estimated claudin protein bands was performed using ImageJ software (National Institutes of Health, Bethesda, MD, U.S.A). Protein concentrations were estimated via Coomassie brilliant blue staining using BSA as the standard.

**Western blotting**. Denatured protein samples were separated via SDS-PAGE using 14% polyacrylamide gels and transferred to polyvinylidene difluoride membranes (Merck, Burlington, MA, USA). The samples were blocked at room temperature for 1 h in PBS containing 0.05% Tween-20 (PBS-T) containing 2% skim milk and incubated with primary antibodies at 4 °C overnight. After washing with PBS-T, the membranes were incubated with HRP-conjugated secondary antibodies at room temperature for 1 h. The bound antibodies were visualised via chemiluminescence imaging using an ImmunoStar kit (Fujifilm Wako Chemicals, Osaka, Japan). The antibodies used for western blotting are listed in Supplementary Table 3.

**Preparation of icHep cell culture system**. Transwell$^{TM}$ permeable support used was coated with 50 μg/ml rat tail collagen I (Corning) in 0.02 N acetic acid at room temperature for 1 h. Transwell$^{TM}$ was washed twice with PBS, then the in vitro-synthesised claudin-1, -2, -3, and -9 proteins were applied to Transwell$^{TM}$ at a density of 750 ng/cm$^2$ each and incubated at room temperature for 1 h. Transwell$^{TM}$ was washed twice with PBS before seeding cells. Cryopreserved human hepatocytes (Donor ID. HU1663) were purchased from Thermo Fisher Scientific and thawed using CHRM. The hepatocytes were then diluted with WEM containing CM3000. PXB-cells (Donor ID.

JFC, and HUM181001B) were obtained from PhoenixBio (Hiroshima, Japan) as described previously[21]. Microscopic observation of hepatocytes treated with 66Z-conjugated magnetic beads revealed an average ratio of human hepatocytes to total freshly isolated PXB-cells of $90.3 \pm 2.9\%$ (22 animals)[21]. PXB-cells were resuspended in 2% DMSO-supplemented hepatocyte growth medium (d-HCGM)[27]. Cryopreserved human hepatocytes and PXB-cells were seeded at a density of $2.1 \times 10^5$ cells/cm$^2$ in the protein-coated culture equipment. After 4 h of incubation, the culture medium was replaced with fresh culture medium to remove the unattached cells. After 24 h of incubation, the cells were cultured in d-HCGM supplemented with 250 μg/ml Matrigel. The spent medium was replaced with fresh medium every 3 days. Donor information for the human hepatocytes used is listed in Supplementary Table 2.

**Counting hepatocyte bile canaliculi opening to the side of the culture equipment**. To count the open- and closed-type bile canaliculi, MRP2 immunostaining was performed using the Z-stack function of a confocal microscope (LSM710; Carl Zeiss, Oberkochen, Germany). Immunostaining was performed as previously described. The ratio of open-type bile canaliculi to the total number of bile canaliculi was calculated according to Eq. (1), and the value obtained was evaluated as the induction rate of open-type bile canaliculi. Here, $A_{open\ bile\ canaliculi}$ and $A_{bile\ canaliculi}$ indicate the number of open-type bile canaliculi and total number of bile canaliculi, respectively.

$$Formation\ rate\,(\%) = \frac{A_{open\ bile\ canaliculi}}{A_{bile\ canaliculi}} \times 100 \qquad (1)$$

**Quantitative evaluation of albumin secretion and urea synthesis**. The level of albumin secretion and urea synthesis in the culture medium accumulated for 3, 5, 7, and 14 days in icHep and sandwich-cultured PXB-cells were measured using sandwich ELISA method and Urea Nitrogen test Wako kit (Fujifilm Wako Chemicals), respectively.

**Quantitative PCR**. Total RNA was extracted from the cell lysates using an RNA iso Plus kit (Takara Bio, Shiga, Japan) according to the manufacturer's instructions. RNA was reverse-transcribed using a high-capacity cDNA reverse transcription kit (Thermo Fisher Scientific). Quantitative PCR was performed using each gene-specific primer (Supplementary Table 4) with a Luna Universal qPCR Master Mix (New England Biolabs, Ipswich, MA, USA).

**Drug uptake assay**. icHep were pre-incubated in transport buffer (TB; 136.7 mM NaCl, 5.36 mM KCl, 25 mM D-glucose, 0.952 mM CaCl$_2$, 0.441 mM KH$_2$PO$_4$, 0.812 mM MgSO$_4$, 0.385 mM Na$_2$HPO$_4$, and 10 mM $N$-(2-hydroxyethyl)-piperazine-$N$-2-(ethanesulphonic acid) [HEPES], adjusted to pH 7.4) at 37°C for 15 min. Drug uptake was initiated by adding TB to the test compounds (1 μCi/ml [$^3$H]E$_2$17βG, [$^3$H]taurocholate, or [$^{14}$C] metformin). Time point of each substrate was set when a linear increase in its intracellular uptake was observed. To terminate the reactions, icHep were rinsed thrice with ice-cold TB at the designated times and lysed in 0.5% Triton X-100/PBS with shaking at room temperature for 1 h. The radioactivity in the cell lysate was measured using a liquid scintillation counter (Accu-FLEX LSC-7200, Hitachi, Tokyo). The amount of hepatocyte protein was determined using a BCA Protein Assay Kit (Fujifilm-Wako Chemicals) with BSA as the standard.

**Metabolic assay**. The metabolic activity of icHep and sandwich-cultured PXB-cells was measured using the cocktail substrate method of drug-metabolizing enzymes. The components of the cocktail are summarized in Supplementary Table 1. Assays were initiated by treating cells with TB containing cocktail substrates. Time point of each substrate was set when a linear increase in its metabolite production was observed. Concentrations of metabolites produced in the medium were measured by liquid chromatography-mass spectrometry (LC-MS/MS).

**Transcellular transport experiments**. Hepatocytes seeded on claudin-coated Transwell™ plates were pre-incubated with TB at 37°C for 15 min. Before the permeation assay, the transepithelial electrical resistance (TEER) of the cells was measured using a MILLICELL-ERS (Merck) as an index of tight junction formation. The permeation assay was initiated by adding TB containing each test compound (25 μM tetramethylrhodamine-dextran [TD4], 5 μM 5-(and-6)-carboxy-2',7'-dichloro-fluorescein diacetate [CDFDA], and 1 μCi/ml [$^3$H]E$_2$17βG, [$^3$H]taurocholate, or [$^3$H]digoxin). To estimate drug biliary clearance ($CL_{bile}$), the assay was initiated by adding TB containing 1 μM of a drug cocktail solution (cimetidine, erythromycin, methotrexate, nafcillin, SN-38, and vincristine) to the cell culture insert (sinusoidal side). The cells were incubated in a 37°C water bath, and 200 μl of TB was collected from the lower chamber (bile side) at 30, 60, 90, and 120 min. An equal amount of TB was immediately added after every sampling point. The concentrations of TD4 and 5-(and-6)-carboxy-2',7'-dichlorofluorescein (CDF) were measured at excitation wavelengths of 550 and 504 nm and emission wavelengths of 572 and 529 nm, respectively using a fluorescent plate reader (1420 ARVO MX/Light, PerkinElmer, Waltham, MA, USA). The radioactivity of [$^3$H]E$_2$17βG, [$^3$H]taurocholate, [$^3$H]digoxin, and [$^{14}$C]metformin was measured using a liquid scintillation counter (AccuFLEX LSC-7200). The concentrations of other drugs were measured using LC-MS/MS, as described below. The apparent permeability coefficient $P_{app}$ (cm/s) was estimated using Eq. (2). Here, $Q$, $A$, and $C_0$ indicate the recovered amount of the compound on the bile side, the surface area of the insert membrane (0.3 cm$^2$), and the initial concentration of the substrate, respectively.

$$P_{app} = dQ/dt/(A \times C_0) \tag{2}$$

The apparent in vitro biliary excretion clearance ($CL_{bile,app}$) of the test compounds obtained via transcellular transport was estimated using Eq. (3).

$$\text{In vitro } CL_{bile,app} = dQ/dt/C_0 \tag{3}$$

In vitro biliary intrinsic clearance ($CL_{bile,int}$) was calculated by subtracting the in vitro $CL_{bile,app}$ of control cells from that of icHep to evaluate the true biliary excretion capacity. In addition, the human in vivo $CL_{bile,int}$ was estimated using Eq. (4), where Qh indicates the human liver blood flow (1400 ml/min/70 kg body weight)[28], and f$_p$ and Rb indicate the blood/plasma concentration ratio and the unbound fraction of plasma proteins of the compound, respectively.

$$CL_{bile,int} = \frac{Qh \times CL_{bile}}{Qh - CL_{bile}/Rb} \times \frac{1}{f_p} \tag{4}$$

The human in vivo $CL_{bile}$, f$_p$, and Rb values for the test compounds are shown in Table 1. The estimated in vitro $CL_{bile,int}$ was scaled up using Eq. (5)[29] to yield the estimated in vivo

$CL_{bile,int}$ (6).

$$SF_{Hepatocyte} = \frac{2160 \times 10^8 \, cells/1800 \, g \, liver/70 \, kg \, human}{0.63 \times 10^5 \, icHep/Transwell} = 3.428 \times 10^6 \tag{5}$$

$$\text{Estimated in vivo } CL_{bile,int} = \text{In vitro } CL_{bile,int} \times SF_{Hepatocyte} \tag{6}$$

The compound concentration in hepatocytes was estimated using Eq. (7)

$$\begin{aligned}&\text{Compound concentration in hepatocytes}\\&= \text{Accumulation in hepatocyte/Intracellular space}\end{aligned} \tag{7}$$

Intracellular space (2.28 μl/10$^6$ hepatocytes) was obtained from reference values[30].

**Evaluation of biliary excretion activity in sandwich-cultured PXB-cells**. The PXB-cells were pre-incubated with TB for 10 min with followed by incubation with TB containing 1 μM of a drug cocktail solution (cimetidine, erythromycin, methotrexate, nafcillin, SN-38, and vincristine) or [$^3$H]digoxin for 2, 5, 10 and 20 min. To disrupt tight junctions, the PXB-cells were pre-incubated with Ca$^{2+}$/Mg$^{2+}$-free TB (containing 1 mM EGTA) for 10 min. The biliary excretion index (BEI) and in vitro $CL_{bile,\,int}$ were estimated using Eqs. (8, 9).

$$BEI = \frac{Accumulation[(+)Ca^{2+}/Mg^{2+}] - Accumulation[(-)Ca^{2+}/Mg^{2+}]}{Accumulation[(+)Ca^{2+}/Mg^{2+}]} \times 100 \tag{8}$$

$$\text{In vitro } CL_{bile,int} = \frac{Accumulation[(+)Ca^{2+}/Mg^{2+}] - Accumulation[(-)Ca^{2+}/Mg^{2+}]}{Incubation \, time \times C_0} \tag{9}$$

**LC-MS/MS analysis**. The concentrations of individual compounds were measured using LC-MS/MS (LCMS-8050, Shimadzu, Kyoto, Japan) connected to an LC-30A system (Shimadzu). The analytical column used was a CAPCELL PAK C18 MGIII (3 μm pore size, ID 2.0 × 50 mm; Osaka Soda, Osaka, Japan) at 40 °C. A 0.1% formic acid aqueous solution (solvent A) and acetonitrile containing 0.1% formic acid (solvent B) were used as the mobile phase. The flow rate was maintained at 0.4 ml/min and the injection volume was 10 μl. The gradient profile was as follows: 5% solvent B for 1.0 min; linear ramp to 95% solvent B for 3.0 min; then return to the initial conditions in 0.5 min. The detection conditions for each compound using LC-MS/MS are summarised in Supplementary Table 5. The lower limit of detection for each compound was 1 nM.

**Statistics and Reproducibility**. Data are presented as the means ± S.D. All experiments described in the main text, the figure legends and Supplementary Information are described with at least three independent replicates. The statistical significance of the differences in each data set was assessed using Student's $t$-test or analysis of variance (ANOVA), followed by the Tukey–Kramer test or Pearson's correlation analysis. Differences were considered statistically significant when the $p$ value was < 0.05.

**Reporting summary**. Further information on research design is available in the Nature Portfolio Reporting Summary linked to this article.

## Data availability

All source data underlying the graphs and charts shown in the main figure have been uploaded as Supplementary Data 1. The images shown in Supplementary Fig. 4 were

cropped from the images provided in Supplementary Fig. 8. All other data are available from the corresponding author (or other sources, as applicable) on reasonable request.

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

## Acknowledgements

This study was supported by the JST Adaptable and Seamless Technology Transfer Program through Target-driven R&D (A-STEP) (JPMJTM19F9, JPMJTM20N9), Grant-in-Aid for Transformative Research Areas (20H05745), AMED (22mk0101249h0001), JSPS KAKENHI Grant Number JP 22J21507, and Kanazawa University SAKIGAKE project 2020. We thank Fujifilm Corporation and PhoenixBio Corporation for providing us with the honeycomb films and PXB-cells used in the study, respectively.

## Author contributions

H.A. and I.T. designed the study. Y.N., N.M. and M.H. conducted the experiments. H.A., Y.N., and I.T. wrote and reviewed the manuscript. H.A., Y.S., A.H. and I.T. supervised the study.

## Competing interests

The authors declare no competing interests.
