## [Peer Review File · Communications Biology]

Reviewers' comments:

Reviewer #1 (Remarks to the Author):

The authors describe a novel in vitro cell model which enables the evaluation of biliary clearance. The manuscript is well written but there are areas for improved data analysis, and discussion that would strengthen the work. Specific comments and recommendations are noted below.

- 1.) The authors should include additional details related to the hepatocytes used in the study, including gender and other demographics (age, viability, plating efficiency, cause of death etc..).
- 2.) Additional markers of hepatic function in this model would be valuable including albumin and urea production for direct comparison with other models including liver MPS.
- 3.) Inclusion of direct comparison with sandwich cultured hepatocytes from the same donor should be considered, including gene expression, enzyme activity and BEI. Evaluation of additional donors in this system would also be valuable (n = 3 for example).
- 4.) Were primary human hepatocytes also included or just PXB hepatocytes? It's unclear the rationale for this and would benefit from a comparison between the two particularly as donor cells in the PXB tend to be juvenile.
- 5.) CYP enzyme activity was investigated at approximately Km concentrations of probe substrate, generally Vmax concentrations should be used to enable comparisons across donors. Was time and concentration linearity evaluated to establish the cocktail assay conditions? Also, mephenytoin is misspelled in the supplement.
- 6.) The authors should comment on why HPRT1 was used as an endogenous control in some assays while ACTB was used in others.
- 7.) Based on Figure 3a/b it appears that the expression of transporters and enzymes rapidly decline between Day 3 and Day 8, it would be beneficial to evaluate days between this time frame to understand the optimal timing for conduct of studies.
- 8.) There is high variability between biological replicates in the derived biliary clearance values, some commentary on what is driving this should be included.
- 9.) The intrinsic clearance values, while well correlated, appear to overestimate the clinical biliary clearance values, some discussion around this observation would be helpful.
- 10.) The authors should consider the impact of bile canalicular flux within the sandwich culture model and this one in the derivation of the biliary clearance.
- 11.) Evaluation of additional human hepatocyte donors may strengthen or change the conclusion related to PXB cells demonstrating higher hemi-lumen formation of bile spaces.
- 12.) In the conclusions three hypothetical scenarios are included without experimental data, one is troglitazone sulfate and hepatotoxicity. Ideally, this could be tested as an example molecule in this system and would enable verification of retention of SULT function and confirmation of utility to test inhibition of biliary efflux transporters. Another example is evaluating the metabolite secretion at the apical and basolateral compartment and lastly, the potential investigation of EHC. It should be noted that microbiome play a significant role in EHC specifically in cleavage of phenolic glucuronides thus, intestinal combo may not be sufficient to assess this potential.

Reviewer #2 (Remarks to the Author):

In this manuscript, a novel hepatocyte model designed for drug biliary excretion capacity assessment is proposed. The authors focused on investigating the contribution of tight junction protein claudins to the formation of bile canaliculi and identified some claudins to mimic hepatobiliary surface. Using this model, the biliary excretion of compounds could be evaluated and the CL_{bile}, int values of several drugs correlated well with results in human according to literature values. The study design is interesting and the results are exciting. However, there are still questions need to be answered.

1. Major comments

- a) According to Fig.2d, it seems that coating with a mixture of claudins (-1, -2, -3, -9) did not

contribute to more hemi-lumen formation in the PXB cells, compared to coating with single claudins. Please explain it. Why didn't the authors choose using single claudins? The selection of claudins was based on the experiments on HepG2 cells that have lower expression of most claudins, did the current claudins combination is the optimized one to induce bile canaliculus formation in different hepatocyte?

b) The resolution of MRP2 staining image in Fig.1d is quite low that the bile canaliculi could not be clearly seen. Please supply images with better resolution. In addition, since the formation of bile canaliculi in icHep is critical to whole work, the author could consider more data to support this, for example, the CDFDA staining, because HepG2 cells alone could expressed MRP2 in some extent.

c) The advance of icHep in pharmacokinetic was mostly characterized by comparing with hepatocyte culture in collagen-coated support. From this perspective, icHep should compare with SCH directly rather than current control (collagen, no claudins) (Fig4., Fig5.). In SCH system, drug transporters expression could be maintained for at least 3 days, while the expression levels of transporters in icHep peaked at day 3.

d) The author should add the prediction results of human in vivo biliary excretion intrinsic clearance estimated from the control (collagen, no claudins) in Table1. From that, we could clearly know the advantage of this novel system in predicting biliary clearance.

e) According to Fig. 5a, the permeability coefficient of nafcillin and SN-38 didn't change after the treatment of inhibitors in icHep control group. Please explain it. In other words, there was no difference in control after inhibitor treatment while huge difference in icHep. Why? Does that came from transporter expression? Did the author measure mRNA or protein levels to verify it?

f) According to Fig. 5b, the CL_{bile}, int of Nafcillin and SN-38 contributed most to the correlation coefficient ($r = 0.998$). The beneath mechanisms should be investigated. Why the compounds in Fig 4 were not included in Fig. 5b, e.g. digoxin, E217βG and taurocholate?

g) The author should indicate the ratio of human-derived hepatocyte in PXB cells, as we know hepatocytes from human liver chimeric mice contain both mouse and human cells. This is important to the correlation of biliary clearance between icHep and human results.

2. Minor comments

a) Fig. 1c, please provide the mRNA or protein expression results of claudins in HepG2 cells after transfected with claudin-expressing plasmids separately.

b) Figure1b, consider creating a segment (break) to Y axis to better compare the expression of claudin-4, -11, -19 between two cells.

c) Page3 Line96, Page6 Line188, please change "increase by" to "increase to".

d) Fig. 3, the figure title is inappropriate since it was the drug uptake assay instead of drug permeation assay.

Response to Reviewers

Re: Manuscript Number: COMMSBIO-22-3410A

Dear Reviewers,

Thank you for constructive comments on our manuscript that improve our study. We carefully considered each comment raised by reviewers and responded to them as followings:

Reviewer #1:

Thank you for very important comments to improve our manuscript. We responded to your comments as followings:

Q1:

The authors should include additional details related to the hepatocytes used in the study, including gender and other demographics (age, viability, plating efficiency, cause of death etc..).

Response:

Thank you for your comment. According to the comment, we have added Supplementary Table 2 on Page 11, Line 1, which includes the demographic information (Donor ID, sex, race, age, and viability) of the hepatocytes used in this study.

Q2:

Additional markers of hepatic function in this model would be valuable including albumin and urea production for direct comparison with other models including liver MPS.

Response:

Thank you for your useful comment. According to your advice, we evaluated albumin secretion and urea synthesis abilities of icHep as indicators of liver function over 14 days of culture by direct comparison with an existing evaluation system, sandwich-cultured PXB-cells. The results of these experiments are described as Figure 3b and 3c. Please refer Figures and Figure legends in the revised manuscript. They are increased linearly with cultivation and are equivalent or higher than the existing sandwich-culture method.

Figure 3b and 3c (Page 20) and Figure legend (Page 22, Line 23-26)

In addition, we have changed the manuscript in Results section as following.

Page 5, Line 145-151, Results:

Accordingly, we selected PXB-cells to construct a permeation assay system and evaluated the usefulness of icHep in evaluating the pharmacokinetic properties of drugs by functional comparison **with conventional sandwich-cultured PXB-cells. First, albumin secretion and urea synthesis were**

measured to assess liver-specific functions of icHep. Albumin secretion and urea synthesis in icHep showed a linear increase throughout the 14-day culture period, and the amounts were significantly higher than those in the sandwich culture system (Fig. 3b, 3c).

Q3:

Inclusion of direct comparison with sandwich cultured hepatocytes from the same donor should be considered, including gene expression, enzyme activity and BEI. Evaluation of additional donors in this system would also be valuable (n = 3 for example).

Response:

Thank you for important comment. According to the comment, we newly prepared icHep from the different liver donor. Initially, although we planned to evaluate using 3 donors, because of the lack of donors, the production of PXB-cells was limited and we were able to evaluate only 2 donors (Donor ID. JFC and HUM181001B). Then, gene expression changes of pharmacokinetic proteins (transporters, drug metabolizing enzymes), drug metabolizing enzyme activities, and biliary excretion activities were directly compared using icHep and SCH derived from donor JFC. In addition, we constructed icHep from donor HUM181001B and compared the icHeps between donor JFC and HUM181001B as well as between icHep and SCH obtaining BEI using PXB-cells from the donor JFC.

Gene expression levels for transporters and drug metabolizing enzymes were comparable between icHep and SCH (Fig. 3d and 3e). Drug metabolizing activity was also comparable between icHep and SCH, while some difference was observed between donors (Fig. 3g and 3h). In addition, effect of transporter inhibitors on permeability of drugs (between Fig. 5a and 5b) and predicted *in vivo* $CL_{bile,int}$ (between Fig. 5d and 5e) were comparable between used two Lot of hepatocytes. Predicted *in vivo* $CL_{bile,int}$ values by BEI obtained from sandwich-cultured hepatocytes was variable depending on incubated time (Fig. 5c and Supplementary Fig. 7).

Figure 3d and 3e (Page 20-21) and Figure legend (Page 22, Line 26-Page 23, Line 3)

Figure 3g and 3h (Page 22) and Figure legend (Page 23, Line 6-15)

Figure 5a-e (Page 26-28) and Figure legend (Page 28, Line 17-31)

Supplementary Figure 7 (Page 8-9) and Figure legend (Page 9, Line 13-19)

In addition, we have changed the manuscript in Results section as below.

Page 5, Line 151-156, Results:

Next, gene expression changes of typical hepatic transporters and drug-metabolizing enzymes during the culture period were evaluated. Most of the investigated gene expression was relatively maintained during the culture period, while gene expression levels of some apical (BSEP) and basal (OATP1B1, OATP1B3) transporters and drug-metabolizing enzymes (CYP1A2, CYP2E1) declined by day 3 of

culture. Each gene expression profile was comparable between icHep and conventional sandwich culture system (Fig 3d, 3e).

Page 5, Line 161-168, Results:

Furthermore, the activity of drug-metabolizing enzymes in icHep and sandwich-cultured PXB-cells obtained through the cocktail substrate method (see Supplementary Table 1) were measured for major CYPs (CYP1A2, CYP2B6, CYP2C9, CYP2C19, CYP2D6, CYP2E1, CYP3A4), and UGT1A1. icHep and sandwich-cultured PXB-cells derived from the same donor exhibited similar metabolic activities, consistent with their gene expression profiles (Fig. 3g). Moreover, donor-to-donor differences (JFC and HUM181001B) were identified in the activities of several drug-metabolizing enzymes (CYP1A2, CYP2C9, CYP2E1, CYP3A4, and UGT1A1) in icHep (Fig. 3g).

Page 6, Line 207-Page 7, Line 227, Results:

We studied the predictability of human *in vivo* biliary excretion using drug permeation assays with icHep from two donors and compared them with conventional sandwich-cultured PXB-cells. Seven drugs whose human *in vivo* biliary excretion clearances are known (cimetidine, digoxin, erythromycin, methotrexate, nafcillin, SN-38, and vincristine) were evaluated. In the permeation assays using icHep, the permeability coefficient of each drug was significantly increased compared to that of the control and decreased in the presence of their respective transporter inhibitors (Fig. 4d, 5a, b, Supplementary Fig. 6). Subsequently, the *in vivo* biliary excretion clearance, plasma protein-unbound fraction, and blood/plasma ratio of each drug were obtained from the literature, and their *in vivo* biliary excretion intrinsic clearances in human were estimated using Equation (4) (see Methods section). The pharmacokinetic and biliary excretion parameters of each drug are summarised in Table 1. The predicted human *in vivo* biliary excretion intrinsic clearance of the drugs in icHep exhibited a good correlation with the corresponding human *in vivo* values [icHep (JFC): $r = 0.969$ and $p = 0.001$, icHep (HUM181001B): $r = 0.966$ and $p = 0.001$] (Fig. 5d, e). Furthermore, the BEI of each drug was calculated as an index of the drug biliary excretion activity using sandwich-cultured PXB-cells for comparison with icHep. However, the calculated BEIs were variable depending on the incubation time of each drug to PXB-cells, and did not show a constant value (Fig. 5c). Furthermore, the prediction accuracy of *in vivo* clearance estimated from BEI in PXB-cells differed depending on the incubation time of each drug (2 min incubation: $r = 0.498$ and $p = 0.255$, 5 min incubation: $r = 0.994$ and $p = 0.001$, 10 min incubation: $r = 0.952$ and $p = 0.001$, 20 min incubation: $r = 0.952$ and $p = 0.001$) (Supplementary Fig. 7). Thus, our icHep permeation assay system efficiently enables more reliable prediction of the human *in vivo* biliary excretion of drugs.

Q4:

Were primary human hepatocytes also included or just PXB hepatocytes? It's unclear the rationale for this and would benefit from a comparison between the two particularly as donor cells in the PXB tend to be juvenile.

Response:

We appreciate the reviewer's comment on this point. We attempted to form a continuous monolayer of primary human hepatocytes derived from human donors on a permeable support. However, primary human hepatocytes on a permeable support formed several cavities (Figure 3a), which prevented us from proceeding to a direct comparative assessment of biliary excretion capacity with icHep using permeation assays. In order to eliminate the occurrence of such cavities, it will be necessary to select hepatocyte lots with excellent cell adhesion and to optimize the culture medium conditions. We think they are next study. This point has been described in the original manuscript.

Figure 3a (Page 20) and Figure legend (Page 22, Line 20-23)

Page 4, Line 139-Page 5, Line 145, Results:

A continuous monolayer of cells with a tight cell-to-cell contact is required for drug permeation assays using cell monolayers. Therefore, we evaluated the morphology of primary human hepatocytes and PXB-cells using a Transwell™ as a permeable support. Primary human hepatocytes we used are difficult to apply for permeability measurements because they form cell-free cavities. In contrast, when icHep was cultured with PXB-cells for 14 days, cavities were observed in the cell monolayer on day 1, polygonal structures characteristic of hepatocytes were observed on day 3, and closely packed hepatocyte monolayers were maintained until day 14 (Fig. 3a).

Q5:

CYP enzyme activity was investigated at approximately K_m concentrations of probe substrate, generally V_{max} concentrations should be used to enable comparisons across donors. Was time and concentration linearity evaluated to establish the cocktail assay conditions? Also, mephenytoin is misspelled in the supplement.

Response:

We thank the reviewer for this comment. In evaluating the drug-metabolizing function of icHep, we chose the cocktail probe method, which is expected to reduce the time, cost and number of samples within limited resources. Substrate concentrations below the K_m value were chosen to primarily activate the CYP pathway of interest and to keep substrate interactions within the cocktail to a minimum considering LC-MS/MS sensitivity. Production rates of substrate metabolites were calculated within the substrate incubation time to ensure linear production of metabolites. Under

established assay conditions, the drug-metabolizing enzymatic activity of icHep constructed from two human liver donor (JFC and HUM181001B)-derived PXB-cells was compared (Figure 3g).

Figure 3g (Page 22) and Figure legend (Page 23, Line 6-10)

In addition, we have changed the manuscript in Methods section as below.

Page 35, Line 6-11, Methods:

The metabolic activity of icHep and sandwich-cultured PXB-cells was measured using the cocktail substrate method of drug-metabolizing enzymes. The components of the cocktail are summarized in Supplementary Table 1. Assays were initiated by treating cells with TB containing cocktail substrates. Time point of each substrate was set when a linear increase in its metabolite production was observed. Concentrations of metabolites produced in the medium were measured by liquid chromatography-mass spectrometry (LC-MS/MS).

Furthermore, the spelling of mephenytoin in Supplementary Table 1 and 5 was properly corrected.

Supplementary Table 1 (Page 10)

Supplementary Table 5 (Page 15)

Q6:

The authors should comment on why HPRT1 was used as an endogenous control in some assays while ACTB was used in others.

Response:

According to the comment, we standardized all endogenous controls used in the assay to *ACTB*. The results of these experiments are described as Figure 1b, 3d, 3e, and Supplementary Figure 2. Please refer Figures and Figure legends in the revised manuscript.

Figure 1b (Page 16) and Figure legend (Page 17, Line 7-11)

Figure 3d-e (Page 20-21) and Figure legend (Page 22, Line 26-Page 23, Line 3)

Supplementary Figure 2 (Page 3) and Figure legend (Page 3, Line 1-5)

Q7:

Based on Figure 3a/b it appears that the expression of transporters and enzymes rapidly decline between Day 3 and Day 8, it would be beneficial to evaluate days between this time frame to understand the optimal timing for conduct of studies.

Response:

Thank you for the important comment. In the current protocol, PXB-cells were obtained seeded in flasks, and then, trypsinized and reseeded onto the permeable support. However, based on the advice from PhoenixBio, the supplier of PXB-cells, we were concerned about the effect of trypsin treatment

on cellular gene expression changes. Therefore, when we implemented a modified protocol in which freshly isolated PXB-cells were seeded on a permeable support, gene expression of transporters and drug-metabolizing enzymes was relatively maintained for 3, 5, 7, and 14 days of culture. Following this result, a modified protocol was used for subsequent cell seeding procedures. The results of these experiments were described as Figure 3d-e. Please refer Figure and Figure legend in the revised manuscript.

Figure 3d-e (Page 20-21) and Figure legend (Page 22, Line 26-Page 23, Line 3)

Page 5, Line 151-156, Results:

Next, gene expression changes of typical hepatic transporters and drug-metabolizing enzymes during the culture period were evaluated. Most of the investigated gene expression was relatively maintained during the culture period, while gene expression levels of some apical (BSEP), and basal (OATP1B1, OATP1B3) transporters and drug-metabolizing enzymes (CYP1A2, CYP2E1) declined by day 3 of culture. Each gene expression profile was comparable between icHep and conventional sandwich culture system (Fig 3d, 3e).

Q8 :

There is high variability between biological replicates in the derived biliary clearance values, some commentary on what is driving this should be included.

Response:

Thank you for thoughtful comment. The *in vitro* biliary excretion intrinsic clearance of icHep was calculated by subtracting that of control from the apparent biliary excretion clearance of icHep obtained by permeation assays. Thus, if the apparent biliary excretion clearance in control and icHep is highly variable between biological replicates, the calculated intrinsic biliary excretion clearance values will be even more variable. This point has been described in the original manuscript as following:

Page 36, Line 6-7, Methods:

In vitro biliary intrinsic clearance ($CL_{bile,int}$) was calculated by subtracting the *in vitro* $CL_{bile,app}$ of control cells from that of icHep to evaluate the true biliary excretion capacity.

In addition, we have changed the manuscript in Discussion section as below.

Page 8, Line 269-272, Discussion:

On the other hand, *in vitro* biliary intrinsic clearance ($CL_{bile,int}$) was calculated by subtracting the *in vitro* $CL_{bile,app}$ of control cells from that of icHep. This subtraction leads to increased variance of the

calculated biliary excretion values. Improvement of formation of open canaliculus is desirable to increase the predictivity.

Q9:

The intrinsic clearance values, while well correlated, appear to overestimate the clinical biliary clearance values, some discussion around this observation would be helpful.

Response:

Thank you for important comments. The predicted *in vivo* intrinsic clearance values by icHep appeared to underestimate the observed *in vivo* values obtained from literatures. One of the factors for this may be the frequency of formation of open bile canaliculus in icHep. The formation rate of open bile canaliculus in icHep is estimated to be 30-40% of the total, and the remaining 60-70% are thought to exist as closed bile canaliculus. Therefore, some of the test substrate accumulates in closed bile canaliculus between hepatocytes, leading to a lower biliary excretion via open bile canaliculus. In addition, there was about 10-fold difference between the NTCP-mediated taurocholate uptake clearance value ($41.3 \pm 8.09 \mu\text{l}/\text{min}/10^6 \text{ cells}$) (Figure 3h) and BSEP-mediated taurocholate biliary excretion clearance value ($3.22 \pm 0.218 \mu\text{l}/\text{min}/10^6 \text{ cells}$) (Figure 4d) of taurocholate in icHep. The rate-limiting process in the hepatobiliary kinetics of the substrate in icHep may be the biliary excretion process. Increasing the formation rate of open bile canaliculus in icHep may be required in eliminating this rate-limiting process.

Figure 3h (Page 22) and Figure legend (Page 23, Line 10-15)

Figure 4d (Page 24) and Figure legend (Page 25, Line 12-19)

Q10:

The authors should consider the impact of bile canalicular flux within the sandwich culture model and this one in the derivation of the biliary clearance.

Response:

Thank you for essential comments. Since the conventional sandwich culture method indirectly estimates the amount of compound accumulated in the bile canaliculus, the prediction of biliary excretion of the compound varies depending on the assay time. It has also been reported that Ca^{2+} depletion in the culture medium causes collapse of the bile canaliculus and decreases NTCP activity [Kumar *et al.*, *AAPS J*, **22**(5):110 (2020)]. In contrast, icHep has an open bile canaliculus and does not require Ca^{2+} depletion to collapse the bile canaliculus, which provides consistent results with no variability over assay time. On the other hand, the icHep system requires subtraction of the apparent biliary excretion clearance calculated under non-claudin-coated control conditions to calculate the biliary excretion intrinsic clearance, which leads to variability. We have added these points to the discussion section.

Page 8, Lines 260-272, Discussion

Although SCHs have the merit of maintaining the expression of drug-metabolizing enzymes and transporters for a long enough time to study drug disposition, they have several drawbacks in evaluating hepatobiliary drug disposition. For example, when the excreted drugs accumulate in the closed lumen, a rapid equilibrium results between the cells and the lumen. **In fact, the calculated BEI and *in vivo* biliary excretion intrinsic clearance of the test drugs varied with drug treatment time in sandwich-cultured PXB-cells (Fig. 5c, Supplementary Fig. 7). Moreover, the depletion of Ca²⁺ in the culture medium affects cell viability and NTCP activity²⁰.** icHep enables the continuous recovery of a drug and its metabolites as they become secreted into the bile side chamber; therefore, they can overcome the issues associated with SCHs and be applied to the robust screening of the biliary excretion of drug candidates. **On the other hand, *In vitro* biliary intrinsic clearance ($CL_{bile,int}$) was calculated by subtracting the *in vitro* $CL_{bile,app}$ of control cells from that of icHep. This subtraction leads to variance of the calculated biliary excretion capacity. Improvement of formation of open canaliculus is desired to increase the predictivity.**

Q11:

Evaluation of additional human hepatocyte donors may strengthen or change the conclusion related to PXB cells demonstrating higher hemi-lumen formation of bile spaces.

Response:

We thank the reviewer for this comment. We considered that the formation of open-form canaliculus in PXB-cells is comparable to that of primary human hepatocyte. We also conducted experiments using additional PXB-cells donor as responded to comment #3 as above. In hemi-lumen formation experiment using PXB-cells derived from two human liver donors (JFC and HUM181001B), the frequency of hemi-lumen formation was similar in both donors. This result indicates that the contribution of claudin to hepatocyte hemi-lumen formation is not limited to a specific hepatocyte donor. The results of these experiments are described as Figure 2c, and Supplementary Figure 5. In addition, we deleted the description about comparison of PXB-cells and primary human hepatocytes. Please refer those Figures and Figure legends in the revised manuscript.

Figure 2c (Page 18) and Figure legend (Page 19, Line 9-11)

Supplementary Figure 5 (Page 6) and Figure legend (Page 6, Line 1-5)

We have changed the manuscript in Results section as below.

Page 4, Line 126-132, Results:

When the icHep was constructed with PXB-cells (donor: JFC), which are fresh hepatocytes derived from human liver chimeric mice¹⁴, on supports individually coated with each claudin, hemi-lumen

formation rates were $44.5 \pm 4.8\%$, $48.9 \pm 11.7\%$, $38.6 \pm 8.1\%$, and $35.9 \pm 6.4\%$ for claudin-1, claudin-2, claudin-3, and claudin-9, respectively (Supplementary Fig. 5). When all these four claudins were coated in a mixture, hemi-lumen formation rates were $49.2 \pm 4.7\%$ (donor: JFC) and $35.8 \pm 6.3\%$ (donor: HUM181001B), respectively. (Supplementary Fig. 5).

Q12:

In the conclusions three hypothetical scenarios are included without experimental data, one is troglitazone sulfate and hepatotoxicity. Ideally, this could be tested as an example molecule in this system and would enable verification of retention of SULT function and confirmation of utility to test inhibition of biliary efflux transporters. Another example is evaluating the metabolite secretion at the apical and basolateral compartment and lastly, the potential investigation of EHC. It should be noted that microbiome play a significant role in EHC specifically in cleavage of phenolic glucuronides thus, intestinal combo may not be sufficient to assess this potential.

Response:

We thank the reviewer for this comment.

12-1: We mentioned troglitazone as an example of involving drug metabolism for BSEP inhibition. However, we considered that the IC_{50} values for troglitazone and its sulfate form for BSEP inhibition (2.3 and 4.2 μ M, respectively) [Pähler *et al.*, *Advances in Molecular Toxicology*, **2**:25-56 (2008)] were too close and to obtain a clear conclusion. We have previously reported using sandwich-cultured human hepatocytes that BSEP is inhibited by candesartan cilexetil (CIL), but not by its active metabolite candesartan (CAN) [Fukuda *et al.*, *Drug Metab Pharmacokinet*, **29**(1):94-6 (2014)]. Therefore, we decided to conduct additional experiment to show the applicability of icHep as an evaluation system for drug-induced cholestasis including drug metabolism using CIL instead of troglitazone. The results of these experiments are described in Figure 6. Please refer Figure and Figure legend in the revised manuscript. In this experiment, we clearly observed BSEP inhibition by CIL but not by CAN, since esterase inhibitor, which inhibits formation of CAN from CIL, increased accumulation of taurocholate in the cells and decreased biliary excretion clearance of taurocholate. So, we deleted discussion about troglitazone and replaced with the result of CAN. Furthermore, we revised manuscript according to the comment.

Figure 6 (Page 29-30) and Figure legend (Page 30, Line 26-36)

We have changed the manuscript in Results and Discussion section as below.

Page 7, Line 229-246, Results:

Evaluation of drug-induced cholestasis (DIC) involving metabolic processes using icHep.

Inhibition of BSEP by drugs and/or their metabolites can lead to DIC with intracellular accumulation of bile acids and subsequent cholestasis, leading to severe liver injury. Since DIC is often associated

with drug metabolism processes, it is desirable to evaluate using SCHs that maintain liver physiology. We have previously reported using SCHs that BSEP is inhibited by candesartan cilexetil (CIL), but not by its active metabolite candesartan (CAN)¹⁸. Here, we investigated the applicability of icHep as an evaluation system for DIC using CIL. Treatment of the cells with CIL alone did not affect biliary excretion clearance of BSEP substrate taurocholate (Fig. 6a). This is due to the rapid hydrolysis of CIL to CAN by esterase in hepatocytes. On the other hand, treatment of the cells with CIL in the presence of esterase inhibitor diisopropyl fluorophosphate (DFP) resulted in the decreased biliary excretion clearance of taurocholate, accompanied by increased intracellular accumulation of taurocholate (Fig. 6a, b). Furthermore, intracellular CIL and CAN were increased and decreased, respectively, by DFP treatment in a concentration dependent manner (Fig. 6c). A significant correlation was observed between the decrease in biliary excretion clearance of taurocholate and the increase in intracellular CIL concentration ($r = -0.984$, $p = 0.01$), whereas there was no correlation with intracellular CAN concentration ($r = 0.274$, $p = 0.66$) (Fig. 6d, e). These results indicate that the icHep permeation assay system can be useful to evaluate DIC, including intrahepatic drug metabolism.

Page 9, Line 290-297, Discussion:

Therefore, we examined the applicability of icHep to the evaluation of biliary excretion inhibition which is affected by intracellular drug metabolism process, using CIL, whose BSEP inhibitory efficacy is attenuated by intracellular metabolism to CAN by esterase. Co-exposure of DFP and CIL decreased biliary clearance of taurocholate with increasing intracellular concentration of CIL. This observation is explained by the inhibition of BSEP-mediated efflux of taurocholate due to decreased intracellular CAN by DFP. Therefore, icHep is applicable for assessment of DIC, including intrahepatic drug metabolism using permeation assays.

12-2: Regarding the assessment of metabolite secretion in the apical and basal compartments, it is difficult at this time to adequately predict substrate distribution in the liver because of concerns about the flux of substrate accumulated in several closed bile ducts in icHep to the sinusoidal medium. Therefore, our claim may have been overstated. To clarify, we have changed the manuscript in Discussion section.

Page 9, Line 305-307, Discussion:

icHep provides easy access to secreted metabolites into the blood and bile side chambers. Therefore, icHep may be used to predict blood levels of metabolites.

12-3: Finally, construction of MPS capable of loading icHep and intestinal cells with microbiome is not currently available, and potential investigation of enterohepatic circulation is a future challenge. We added description about intestinal microbiome.

Page 9, Line 307-313, Discussion:

In addition, biliary-excreted parental drugs and metabolites are reabsorbed in the gastrointestinal tract, in a process called enterohepatic circulation. It has a significant impact on drug pharmacokinetics, efficacy, and safety. MPS is useful for studying inter-organ relationships; however, enterohepatic circulation is difficult to establish using currently available cell culture methods. It is expected that MPS, including organ-on-a-chip systems equipped with icHep, intestinal cells, and intestinal microbiome will enable to evaluate complicated enterohepatic circulation.

Reviewer #2:

We wish to express our appreciation to the reviewer for your insightful comments on our paper. We responded to your comments as followings:

Major comments

Q1:

According to Fig.2d, it seems that coating with a mixture of claudins (-1, -2, -3, -9) did not contribute to more hemi-lumen formation in the PXB cells, compared to coating with single claudins. Please explain it. Why didn't the authors choose using single claudins? The selection of claudins was based on the experiments on HepG2 cells that have lower expression of most claudins, did the current claudins combination is the optimized one to induce bile canaliculus formation in different hepatocyte?

Response:

We thank the reviewer for this comment. Claudins can trans-interact with heterologous claudin molecules expressed on neighboring cell membranes [Furuse *et al.*, *J Cell Biol*, **147**(4):891-903 (1999)]. We used claudin-1, -2, -3, and -9 mixture rather than single claudin because it means that the more diverse the claudin molecules used in the coating, the higher the probability of device-cell claudin interactions. As pointed out by the reviewer, it may be possible to construct an evaluation system with higher functionality by screening the optimal claudin molecules and combinations for human hepatocytes. However, the most important point of our study is to show that the bile secretion process can be evaluated by permeation assay using claudin-coated plate. We would like to set studies of claudin optimization as future work. In response to reviewer's comment, the manuscript was modified as followings.

Figure 1b (Page 16) and Figure legend (Page 17, Line 7-11)

Supplementary Figure 5 (Page 6) and Figure legend (Page 6, Line 1-5)

We added the following sentences to the Discussion section.

Page 8, Line 275-279, Discussion:

On the other hand, claudin was narrowed down using HepG2 cells (Fig. 1c). Since claudin genes expressed in HepG2 cells have different patterns from human hepatocytes (Fig. 1b), it is possible to construct icHep with higher formation of open-form canaliculus by further optimizing the types and combinations of claudin proteins using human hepatocytes.

Q2:

The resolution of MRP2 staining image in Fig.1d is quite low that the bile canaliculi could not be clearly seen. Please supply images with better resolution. In addition, since the formation of bile canaliculi in icHep is critical to whole work, the author could consider more data to support this, for example, the CDFDA staining, because HepG2 cells alone could expressed MRP2 in some extent.

Response:

We appreciate the reviewer's comment on this point. According to the comment, we newly acquired a high-resolution bile canaliculus image.

Figure 1d (Page 17) and Figure legend (Page 17, Line 16-20)

We used HepG2 cells in Figure 1d for initial confirmation of the concept that bile canaliculus localization is induced in the direction of claudin action. On the other hand, the bile canaliculus of HepG2 cells exhibited an immature spherical morphology, making them not suitable tools for assessing the formation of open hemi-lumens. Therefore, the formation of open hemi-lumens by claudin coating was evaluated by MRP2 immunostaining using primary human hepatocytes (Figure 2b) and PXB-cells (Figure 3f) that form more physiological bile canaliculus. In addition, a CDFDA permeation assay using icHep was performed, and the MRP2-mediated transport of CDF to the biliary side (lower chamber) in icHep was significantly higher than that in PXB-cells cultured on a collagen-coated permeable support (Figure 4c). These result supports the biliary excretion of CDF through the open hemi-lumen in icHep.

Figure 2b (Page 18) and Figure legend (Page 19, Line 3-9)

Figure 3f (Page 21) and Figure legend (Page 23, Line 3-6)

Figure 4c (Page 24) and Figure legend (Page 25, Line 7-12)

Q3:

The advance of icHep in pharmacokinetic was mostly characterized by comparing with hepatocyte culture in collagen-coated support. From this perspective, icHep should compare with SCH directly rather than current control (collagen, no claudins) (Fig4., Fig5.). In SCH system, drug transporters

expression could be maintained for at least 3 days, while the expression levels of transporters in icHep peaked at day 3.

Response:

Thank you for thoughtful comment. According to the comment, we conducted experiments, including gene expression changes of pharmacokinetic proteins (transporters and drug metabolizing enzymes), drug metabolizing enzyme activities, and biliary excretion activities, to compare icHep and sandwich-cultured PXB-cells (SCH) derived from donor JFC. In the current protocol, PXB-cells were obtained seeded in flasks, trypsinized and reseeded onto the permeable support. However, based on advice from PhoenixBio, the supplier of PXB-cells, we were concerned about the effect of trypsin treatment on cellular gene expression changes. Therefore, when we implemented a modified protocol in which freshly isolated PXB-cells were seeded on a permeable support, gene expression of transporters and drug-metabolizing enzymes was relatively maintained for 3, 5, 7, and 14 days of culture. Gene expression levels for transporters and drug metabolizing enzymes were comparable between icHep and SCH (Fig. 3d and 3e). Drug metabolizing activity was also comparable between icHep and SCH, while some difference was observed between donors (Fig. 3g and 3h). In addition, effect of transporter inhibitors on permeability of drugs (between Fig. 5a and 5b) and predicted *in vivo* $CL_{bile,int}$ (between Fig. 5d and 5e) were comparable between used two Lot of hepatocytes. Predicted *in vivo* $CL_{bile,int}$ values by BEI was variable depending on incubated time (Fig. 5c and Supplementary Fig. 7).

Figure 3d and 3e (Page 20-21) and Figure legend (Page 22, Line 26-Page 23, Line 3)

Figure 3g and 3h (Page 22) and Figure legend (Page 23, Line 6-15)

Figure 5a-e (Page 26-28) and Figure legend (Page 28, Line 17-31)

Supplementary Figure 7 (Page 8-9) and Figure legend (Page 9, Line 13-19)

We have changed the manuscript in Results section as below.

Page 5, Line 151-156, Results:

Next, gene expression changes of typical hepatic transporters and drug-metabolizing enzymes during the culture period were evaluated. Most of the investigated gene expression was relatively maintained during the culture period, while gene expression levels of some apical (BSEP), and basal (OATP1B1, OATP1B3) transporters and drug-metabolizing enzymes (CYP1A2, CYP2E1) declined by day 3 of culture. Each gene expression profile was comparable between icHep and conventional sandwich culture system (Fig 3d, 3e).

Page 5, Line 161-168, Results:

Furthermore, the activity of drug-metabolizing enzymes in icHep and sandwich-cultured PXB-cells obtained through the cocktail substrate method (see Supplementary Table 1) were measured for major CYPs (CYP1A2, CYP2B6, CYP2C9, CYP2C19, CYP2D6, CYP2E1, CYP3A4), and UGT1A1. icHep and sandwich-cultured PXB-cells derived from the same donor exhibited similar metabolic activities,

consistent with their gene expression profiles (Fig. 3g). Moreover, donor-to-donor differences (JFC and HUM181001B) were identified in the activities of several drug-metabolizing enzymes (CYP1A2, CYP2C9, CYP2E1, CYP3A4, and UGT1A1) in icHep (Fig. 3g).

Page 6, Line 207-Page 7, Line 227, Results:

We studied the predictability of human *in vivo* biliary excretion using drug permeation assays with icHep from two donors and compared them with conventional sandwich-cultured PXB-cells. Seven drugs whose human *in vivo* biliary excretion clearances are known (cimetidine, digoxin, erythromycin, methotrexate, nafcillin, SN-38, and vincristine) were evaluated. In the permeation assays using icHep, the permeability coefficient of each drug was significantly increased compared to that of the control and decreased in the presence of their respective transporter inhibitors (Fig. 4d, 5a, b, Supplementary Fig. 6). Subsequently, the *in vivo* biliary excretion clearance, plasma protein-unbound fraction, and blood/plasma ratio of each drug were obtained from the literature, and their *in vivo* biliary excretion intrinsic clearances in human were estimated using Equation (4) (see Methods section). The pharmacokinetic and biliary excretion parameters of each drug are summarised in Table 1. The predicted human *in vivo* biliary excretion intrinsic clearance of the drugs in icHep exhibited a good correlation with the corresponding human *in vivo* values [icHep (JFC): $r = 0.969$ and $p = 0.001$, icHep (HUM181001B): $r = 0.966$ and $p = 0.001$] (Fig. 5d, e). Furthermore, the BEI of each drug was calculated as an index of the drug biliary excretion activity using sandwich-cultured PXB-cells for comparison with icHep. However, the calculated BEIs were variable depending on the incubation time of each drug to PXB-cells, and did not show a constant value (Fig. 5c). Furthermore, the prediction accuracy of *in vivo* clearance estimated from BEI in PXB-cells differed depending on the incubation time of each drug (2 min incubation: $r = 0.498$ and $p = 0.255$, 5 min incubation: $r = 0.994$ and $p = 0.001$, 10 min incubation: $r = 0.952$ and $p = 0.001$, 20 min incubation: $r = 0.952$ and $p = 0.001$) (Supplementary Fig. 7). Thus, our icHep permeation assay system efficiently enables more reliable prediction of the human *in vivo* biliary excretion of drugs.

Q4:

The author should add the prediction results of human *in vivo* biliary excretion intrinsic clearance estimated from the control (collagen, no claudins) in Table 1. From that, we could clearly know the advantage of this novel system in predicting biliary clearance.

Response:

Thank you for the comment. The *in vitro* biliary excretion intrinsic clearance of icHep was calculated by subtracting that of control from the apparent biliary excretion clearance of icHep obtained by permeation assays. Therefore, *in vitro* and *in vivo* biliary excretion intrinsic clearance from control conditions could not be calculated separately. Instead, we conducted the conventional SCH method to predict bile secretion to compare with icHep. We considered that icHep will enable stable evaluation,

because conventional SCH method yields different results depending on the time point. This point has been described in the original manuscript as following:

Page 7, Lines 219-227, Results:

Furthermore, the BEI of each drug was calculated as an index of the drug biliary excretion activity using sandwich-cultured PXB-cells for comparison with icHep. However, the calculated BEIs were variable depending on the incubation time of each drug to PXB-cells, and did not show a constant value (Fig. 5c). In addition, the prediction accuracy of *in vivo* clearance estimated from BEI in PXB-cells differed depending on the incubation time of each drug (2 min incubation: $r = 0.498$ and $p = 0.255$, 5 min incubation: $r = 0.994$ and $p = 0.001$, 10 min incubation: $r = 0.952$ and $p = 0.001$, 20 min incubation: $r = 0.952$ and $p = 0.001$) (Supplementary Fig. 7). Thus, our icHep permeation assay system efficiently enables more reliable prediction of the human *in vivo* biliary excretion of drugs.

Page 36, Line 6-7, Methods:

In vitro biliary intrinsic clearance ($CL_{bile,int}$) was calculated by subtracting the *in vitro* $CL_{bile,app}$ of control cells from that of icHep to evaluate the true biliary excretion capacity.

Q5:

According to Fig. 5a, the permeability coefficient of nafcillin and SN-38 didn't change after the treatment of inhibitors in icHep control group. Please explain it. In other words, there was no difference in control after inhibitor treatment while huge difference in icHep. Why? Does that come from transporter expression? Did the author measure mRNA or protein levels to verify it?

Response:

Thank you for the comment. PXB-cells cultured on a collagen-coated permeable support (control condition) form closed bile canaliculus between adjacent hepatocytes. Thus, biliary excreted drug accumulates in the closed bile canaliculus, resulting in no biliary transporter-mediated transport of substrate to the lower chamber (bile side). Accordingly, the absence of change in substrate permeability by transporter inhibitor treatment is the expected result.

Q6:

According to Fig. 5b, the $CL_{bile, int}$ of Nafcillin and SN-38 contributed most to the correlation coefficient ($r = 0.998$). The beneath mechanisms should be investigated. Why the compounds in Fig 4 were not included in Fig. 5b, e.g. digoxin, E217 β G and taurocholate?

Response:

We appreciate the reviewer's comment. We understand the comment by the reviewer but this is due to the very limited previous reports on the *in vivo* biliary excretion clearance in human. Among the

compounds investigated in Figure 4, digoxin, E₂17βG, and taurocholate, we found the information on human *in vivo* biliary excretion clearance only for digoxin. Therefore, the predicted human *in vivo* biliary excretion intrinsic clearance of digoxin was calculated and subjected to correlation analysis with observed values in human (Figure 5d, 5e, Supplementary Figure 7, and Table 1). On the other hand, human *in vivo* clearance information for E₂17βG and taurocholate was not found as far as we investigated, so they were not included in the correlation diagram.

We understand that the intrinsic biliary excretion clearance of SN-38 and Nafcillin contributed most to the correlation coefficient. We calculated the intrinsic clearance in human *in vivo* biliary excretion according to Equation (4) in the Methods section. Due to the presence of the plasma unbound fraction (f_p) term in the denominator of Equation (4), intrinsic clearance values are significantly affected by protein binding of the test compound. The plasma unbound fraction, f_p values of SN-38 and Nafcillin are 0.00887 and 0.123, respectively (Table 1). This value is the lowest and second lowest among the tested compounds. Therefore, the high intrinsic clearance values of SN-38 and Nafcillin may be attributed to their high protein binding properties.

Figure 5d-e (Page 27-28) and Figure legend (Page 28, Line 28-29)

Supplementary Figure 7 (Page 8-9) and Figure legend (Page 9, Line 13-19)

Table 1 (Page 11)

Q7:

The author should indicate the ratio of human-derived hepatocyte in PXB cells, as we know hepatocytes from human liver chimeric mice contain both mouse and human cells. This is important to the correlation of biliary clearance between icHep and human results.

Response:

Thank you for thoughtful comment. Microscopic observation of hepatocytes treated with 66Z-conjugated magnetic beads was performed to confirm the ratio of human hepatocytes in PXB-cells and confirmed that the ratio of human hepatocytes was $90.3 \pm 2.9\%$ (22 animals) [Yamasaki *et al.*, *PLOS One*, **15**(9):e0237809 (2020)]. This indicates that about 10% of mouse hepatocytes are present in PXB-cells, but almost all hepatocytes are replaced with human type.

We have changed the manuscript in Discussion and Methods section as below.

Page 8, Line 272-275, Discussion:

In addition, PXB-cells used for constructing icHep contain about 10% mouse-derived cells²¹. Therefore, some of the obtained biliary transport includes mouse-derived reactions. Further study is required to establish icHep with pure human hepatocytes such as improved primary cultured hepatocytes^{22,23} and iPS-derived hepatocytes²⁴.

Page 33, Line 33-35, Methods:

Microscopic observation of hepatocytes treated with 66Z-conjugated magnetic beads revealed an average ratio of human hepatocytes to total freshly isolated PXB-cells of $90.3 \pm 2.9\%$ (22 animals)²¹.

Minor comments

Q1:

Fig. 1c, please provide the mRNA or protein expression results of claudins in HepG2 cells after transfected with claudin-expressing plasmids separately.

Response:

Thank you for thoughtful comment. As pointed out by the reviewer, claudin mRNA expression analysis was performed after individual transfection of claudin plasmids. Gene expression of each claudin increased in the plasmid-introduced group compared to the control group (Supplementary Figure 2).

Supplementary Figure 2 (Page 3) and Figure legend (Page 3, Line 1-5)

We have changed the manuscript in Results section as below.

Page 3, Line 94-95, Results:

All claudin molecules investigated were efficiently transfected into HepG2 cells by the lentiviral method (Supplementary Fig. 2).

Q2:

Figure1b, consider creating a segment (break) to Y axis to better compare the expression of claudin-4, -11, -19 between two cells.

Response:

Thank you for your comments. Separate graphs have been added for some claudin molecules (CLDN8, 17, 18, 19, 25 and 27) to facilitate comparison of claudin expression between the HepG2 cells and human liver in Fig 1b.

Figure 1b (Page 16) and Figure legend (Page 17, Line 7-11)

Q3:

Page3 Line96, Page6 Line188, please change “increase by” to “increase to”.

Response:

Thank you for your comments. The relevant parts of the manuscript have been corrected as appropriate.

Page 3, Line 95-97, Results:

The expression of claudin-1, -2, -3, and -9 **increased** the number of bile canaliculi in HepG2 cells to 150 ± 14 , 158 ± 15 , 145 ± 21 , and $142 \pm 15\%$, respectively, compared to mock cells (Fig. 1c).

Page 6, Line 193-195, Results:

When CDFDA was loaded onto the sinusoidal side (upper chamber), the appearance of CDF on the bile side (lower chamber) of icHep **increased to** $147 \pm 14.6\%$ of the control group.

Q4:

Fig. 3, the figure title is inappropriate since it was the drug uptake assay instead of drug permeation assay.

Response:

Thank you for your comments. Figure 3 legend title changed to: **Pharmacokinetic functional evaluation of icHep.**

REVIEWERS' COMMENTS:

Reviewer #1 (Remarks to the Author):

Thank you to the authors for considering the feedback provided. All of my concerns have been adequately addressed and I feel that the manuscript is much improved.

Reviewer #2 (Remarks to the Author):

The revised version addressed most of the concerns the reviewer pointed out before. The reviewer doesn't have further questions.

Response to Referees

Re: Manuscript Number: COMMSBIO-22-3410A

Dear Reviewers,

Thank you for constructive comments on our manuscript that improve our study. We carefully considered each comment raised by reviewers and responded to them as followings:

Reviewer #1:

Thank you for very important comments to improve our manuscript. We responded to your comments as followings:

Q1:

The authors should include additional details related to the hepatocytes used in the study, including gender and other demographics (age, viability, plating efficiency, cause of death etc..).

Response:

Thank you for your comment. According to the comment, we have added Supplementary Table 2 on Page 11, Line 1, which includes the demographic information (Donor ID, sex, race, age, and viability) of the hepatocytes used in this study.

Q2:

Additional markers of hepatic function in this model would be valuable including albumin and urea production for direct comparison with other models including liver MPS.

Response:

Thank you for your useful comment. According to your advice, we evaluated albumin secretion and urea synthesis abilities of icHep as indicators of liver function over 14 days of culture by direct comparison with an existing evaluation system, sandwich-cultured PXB-cells. The results of these experiments are described as Figure 3b and 3c. Please refer Figures and Figure legends in the revised manuscript. They are increased linearly with cultivation and are equivalent or higher than the existing sandwich-culture method.

Figure 3b and 3c (Page 20) and Figure legend (Page 22, Line 23-26)

In addition, we have changed the manuscript in Results section as following.

Page 5, Line 145-151, Results:

Accordingly, we selected PXB-cells to construct a permeation assay system and evaluated the usefulness of icHep in evaluating the pharmacokinetic properties of drugs by functional comparison **with conventional sandwich-cultured PXB-cells. First, albumin secretion and urea synthesis were**

measured to assess liver-specific functions of icHep. Albumin secretion and urea synthesis in icHep showed a linear increase throughout the 14-day culture period, and the amounts were significantly higher than those in the sandwich culture system (Fig. 3b, 3c).

Q3:

Inclusion of direct comparison with sandwich cultured hepatocytes from the same donor should be considered, including gene expression, enzyme activity and BEI. Evaluation of additional donors in this system would also be valuable (n = 3 for example).

Response:

Thank you for important comment. According to the comment, we newly prepared icHep from the different liver donor. Initially, although we planned to evaluate using 3 donors, because of the lack of donors, the production of PXB-cells was limited and we were able to evaluate only 2 donors (Donor ID. JFC and HUM181001B). Then, gene expression changes of pharmacokinetic proteins (transporters, drug metabolizing enzymes), drug metabolizing enzyme activities, and biliary excretion activities were directly compared using icHep and SCH derived from donor JFC. In addition, we constructed icHep from donor HUM181001B and compared the icHeps between donor JFC and HUM181001B as well as between icHep and SCH obtaining BEI using PXB-cells from the donor JFC.

Gene expression levels for transporters and drug metabolizing enzymes were comparable between icHep and SCH (Fig. 3d and 3e). Drug metabolizing activity was also comparable between icHep and SCH, while some difference was observed between donors (Fig. 3g and 3h). In addition, effect of transporter inhibitors on permeability of drugs (between Fig. 5a and 5b) and predicted *in vivo* $CL_{bile,int}$ (between Fig. 5d and 5e) were comparable between used two Lot of hepatocytes. Predicted *in vivo* $CL_{bile,int}$ values by BEI obtained from sandwich-cultured hepatocytes was variable depending on incubated time (Fig. 5c and Supplementary Fig. 7).

Figure 3d and 3e (Page 20-21) and Figure legend (Page 22, Line 26-Page 23, Line 3)

Figure 3g and 3h (Page 22) and Figure legend (Page 23, Line 6-15)

Figure 5a-e (Page 26-28) and Figure legend (Page 28, Line 17-31)

Supplementary Figure 7 (Page 8-9) and Figure legend (Page 9, Line 13-19)

In addition, we have changed the manuscript in Results section as below.

Page 5, Line 151-156, Results:

Next, gene expression changes of typical hepatic transporters and drug-metabolizing enzymes during the culture period were evaluated. Most of the investigated gene expression was relatively maintained during the culture period, while gene expression levels of some apical (BSEP) and basal (OATP1B1, OATP1B3) transporters and drug-metabolizing enzymes (CYP1A2, CYP2E1) declined by day 3 of

culture. Each gene expression profile was comparable between icHep and conventional sandwich culture system (Fig 3d, 3e).

Page 5, Line 161-168, Results:

Furthermore, the activity of drug-metabolizing enzymes in icHep and sandwich-cultured PXB-cells obtained through the cocktail substrate method (see Supplementary Table 1) were measured for major CYPs (CYP1A2, CYP2B6, CYP2C9, CYP2C19, CYP2D6, CYP2E1, CYP3A4), and UGT1A1. icHep and sandwich-cultured PXB-cells derived from the same donor exhibited similar metabolic activities, consistent with their gene expression profiles (Fig. 3g). Moreover, donor-to-donor differences (JFC and HUM181001B) were identified in the activities of several drug-metabolizing enzymes (CYP1A2, CYP2C9, CYP2E1, CYP3A4, and UGT1A1) in icHep (Fig. 3g).

Page 6, Line 207-Page 7, Line 227, Results:

We studied the predictability of human *in vivo* biliary excretion using drug permeation assays with icHep from two donors and compared them with conventional sandwich-cultured PXB-cells. Seven drugs whose human *in vivo* biliary excretion clearances are known (cimetidine, digoxin, erythromycin, methotrexate, nafcillin, SN-38, and vincristine) were evaluated. In the permeation assays using icHep, the permeability coefficient of each drug was significantly increased compared to that of the control and decreased in the presence of their respective transporter inhibitors (Fig. 4d, 5a, b, Supplementary Fig. 6). Subsequently, the *in vivo* biliary excretion clearance, plasma protein-unbound fraction, and blood/plasma ratio of each drug were obtained from the literature, and their *in vivo* biliary excretion intrinsic clearances in human were estimated using Equation (4) (see Methods section). The pharmacokinetic and biliary excretion parameters of each drug are summarised in Table 1. The predicted human *in vivo* biliary excretion intrinsic clearance of the drugs in icHep exhibited a good correlation with the corresponding human *in vivo* values [icHep (JFC): $r = 0.969$ and $p = 0.001$, icHep (HUM181001B): $r = 0.966$ and $p = 0.001$] (Fig. 5d, e). Furthermore, the BEI of each drug was calculated as an index of the drug biliary excretion activity using sandwich-cultured PXB-cells for comparison with icHep. However, the calculated BEIs were variable depending on the incubation time of each drug to PXB-cells, and did not show a constant value (Fig. 5c). Furthermore, the prediction accuracy of *in vivo* clearance estimated from BEI in PXB-cells differed depending on the incubation time of each drug (2 min incubation: $r = 0.498$ and $p = 0.255$, 5 min incubation: $r = 0.994$ and $p = 0.001$, 10 min incubation: $r = 0.952$ and $p = 0.001$, 20 min incubation: $r = 0.952$ and $p = 0.001$) (Supplementary Fig. 7). Thus, our icHep permeation assay system efficiently enables more reliable prediction of the human *in vivo* biliary excretion of drugs.

Q4:

Were primary human hepatocytes also included or just PXB hepatocytes? It's unclear the rationale for this and would benefit from a comparison between the two particularly as donor cells in the PXB tend to be juvenile.

Response:

We appreciate the reviewer's comment on this point. We attempted to form a continuous monolayer of primary human hepatocytes derived from human donors on a permeable support. However, primary human hepatocytes on a permeable support formed several cavities (Figure 3a), which prevented us from proceeding to a direct comparative assessment of biliary excretion capacity with icHep using permeation assays. In order to eliminate the occurrence of such cavities, it will be necessary to select hepatocyte lots with excellent cell adhesion and to optimize the culture medium conditions. We think they are next study. This point has been described in the original manuscript.

Figure 3a (Page 20) and Figure legend (Page 22, Line 20-23)

Page 4, Line 139-Page 5, Line 145, Results:

A continuous monolayer of cells with a tight cell-to-cell contact is required for drug permeation assays using cell monolayers. Therefore, we evaluated the morphology of primary human hepatocytes and PXB-cells using a Transwell™ as a permeable support. Primary human hepatocytes we used are difficult to apply for permeability measurements because they form cell-free cavities. In contrast, when icHep was cultured with PXB-cells for 14 days, cavities were observed in the cell monolayer on day 1, polygonal structures characteristic of hepatocytes were observed on day 3, and closely packed hepatocyte monolayers were maintained until day 14 (Fig. 3a).

Q5:

CYP enzyme activity was investigated at approximately K_m concentrations of probe substrate, generally V_{max} concentrations should be used to enable comparisons across donors. Was time and concentration linearity evaluated to establish the cocktail assay conditions? Also, mephenytoin is misspelled in the supplement.

Response:

We thank the reviewer for this comment. In evaluating the drug-metabolizing function of icHep, we chose the cocktail probe method, which is expected to reduce the time, cost and number of samples within limited resources. Substrate concentrations below the K_m value were chosen to primarily activate the CYP pathway of interest and to keep substrate interactions within the cocktail to a minimum considering LC-MS/MS sensitivity. Production rates of substrate metabolites were calculated within the substrate incubation time to ensure linear production of metabolites. Under

established assay conditions, the drug-metabolizing enzymatic activity of icHep constructed from two human liver donor (JFC and HUM181001B)-derived PXB-cells was compared (Figure 3g).

Figure 3g (Page 22) and Figure legend (Page 23, Line 6-10)

In addition, we have changed the manuscript in Methods section as below.

Page 35, Line 6-11, Methods:

The metabolic activity of icHep and sandwich-cultured PXB-cells was measured using the cocktail substrate method of drug-metabolizing enzymes. The components of the cocktail are summarized in Supplementary Table 1. Assays were initiated by treating cells with TB containing cocktail substrates. Time point of each substrate was set when a linear increase in its metabolite production was observed. Concentrations of metabolites produced in the medium were measured by liquid chromatography-mass spectrometry (LC-MS/MS).

Furthermore, the spelling of mephenytoin in Supplementary Table 1 and 5 was properly corrected.

Supplementary Table 1 (Page 10)

Supplementary Table 5 (Page 15)

Q6:

The authors should comment on why HPRT1 was used as an endogenous control in some assays while ACTB was used in others.

Response:

According to the comment, we standardized all endogenous controls used in the assay to *ACTB*. The results of these experiments are described as Figure 1b, 3d, 3e, and Supplementary Figure 2. Please refer Figures and Figure legends in the revised manuscript.

Figure 1b (Page 16) and Figure legend (Page 17, Line 7-11)

Figure 3d-e (Page 20-21) and Figure legend (Page 22, Line 26-Page 23, Line 3)

Supplementary Figure 2 (Page 3) and Figure legend (Page 3, Line 1-5)

Q7:

Based on Figure 3a/b it appears that the expression of transporters and enzymes rapidly decline between Day 3 and Day 8, it would be beneficial to evaluate days between this time frame to understand the optimal timing for conduct of studies.

Response:

Thank you for the important comment. In the current protocol, PXB-cells were obtained seeded in flasks, and then, trypsinized and reseeded onto the permeable support. However, based on the advice from PhoenixBio, the supplier of PXB-cells, we were concerned about the effect of trypsin treatment

on cellular gene expression changes. Therefore, when we implemented a modified protocol in which freshly isolated PXB-cells were seeded on a permeable support, gene expression of transporters and drug-metabolizing enzymes was relatively maintained for 3, 5, 7, and 14 days of culture. Following this result, a modified protocol was used for subsequent cell seeding procedures. The results of these experiments were described as Figure 3d-e. Please refer Figure and Figure legend in the revised manuscript.

Figure 3d-e (Page 20-21) and Figure legend (Page 22, Line 26-Page 23, Line 3)

Page 5, Line 151-156, Results:

Next, gene expression changes of typical hepatic transporters and drug-metabolizing enzymes during the culture period were evaluated. Most of the investigated gene expression was relatively maintained during the culture period, while gene expression levels of some apical (BSEP), and basal (OATP1B1, OATP1B3) transporters and drug-metabolizing enzymes (CYP1A2, CYP2E1) declined by day 3 of culture. Each gene expression profile was comparable between icHep and conventional sandwich culture system (Fig 3d, 3e).

Q8 :

There is high variability between biological replicates in the derived biliary clearance values, some commentary on what is driving this should be included.

Response:

Thank you for thoughtful comment. The *in vitro* biliary excretion intrinsic clearance of icHep was calculated by subtracting that of control from the apparent biliary excretion clearance of icHep obtained by permeation assays. Thus, if the apparent biliary excretion clearance in control and icHep is highly variable between biological replicates, the calculated intrinsic biliary excretion clearance values will be even more variable. This point has been described in the original manuscript as following:

Page 36, Line 6-7, Methods:

In vitro biliary intrinsic clearance ($CL_{bile,int}$) was calculated by subtracting the *in vitro* $CL_{bile,app}$ of control cells from that of icHep to evaluate the true biliary excretion capacity.

In addition, we have changed the manuscript in Discussion section as below.

Page 8, Line 269-272, Discussion:

On the other hand, *in vitro* biliary intrinsic clearance ($CL_{bile,int}$) was calculated by subtracting the *in vitro* $CL_{bile,app}$ of control cells from that of icHep. This subtraction leads to increased variance of the calculated biliary excretion values. Improvement of formation of open canaliculus is desirable to increase the predictivity.

Q9:

The intrinsic clearance values, while well correlated, appear to overestimate the clinical biliary clearance values, some discussion around this observation would be helpful.

Response:

Thank you for important comments. The predicted *in vivo* intrinsic clearance values by icHep appeared to underestimate the observed *in vivo* values obtained from literatures. One of the factors for this may be the frequency of formation of open bile canaliculus in icHep. The formation rate of open bile canaliculus in icHep is estimated to be 30-40% of the total, and the remaining 60-70% are thought to exist as closed bile canaliculus. Therefore, some of the test substrate accumulates in closed bile canaliculus between hepatocytes, leading to a lower biliary excretion via open bile canaliculus. In addition, there was about 10-fold difference between the NTCP-mediated taurocholate uptake clearance value ($41.3 \pm 8.09 \mu\text{l}/\text{min}/10^6 \text{ cells}$) (Figure 3h) and BSEP-mediated taurocholate biliary excretion clearance value ($3.22 \pm 0.218 \mu\text{l}/\text{min}/10^6 \text{ cells}$) (Figure 4d) of taurocholate in icHep. The rate-limiting process in the hepatobiliary kinetics of the substrate in icHep may be the biliary excretion process. Increasing the formation rate of open bile canaliculus in icHep may be required in eliminating this rate-limiting process.

Figure 3h (Page 22) and Figure legend (Page 23, Line 10-15)

Figure 4d (Page 24) and Figure legend (Page 25, Line 12-19)

Q10:

The authors should consider the impact of bile canalicular flux within the sandwich culture model and this one in the derivation of the biliary clearance.

Response:

Thank you for essential comments. Since the conventional sandwich culture method indirectly estimates the amount of compound accumulated in the bile canaliculus, the prediction of biliary excretion of the compound varies depending on the assay time. It has also been reported that Ca^{2+} depletion in the culture medium causes collapse of the bile canaliculus and decreases NTCP activity [Kumar *et al.*, *AAPS J*, **22**(5):110 (2020)]. In contrast, icHep has an open bile canaliculus and does not require Ca^{2+} depletion to collapse the bile canaliculus, which provides consistent results with no variability over assay time. On the other hand, the icHep system requires subtraction of the apparent biliary excretion clearance calculated under non-claudin-coated control conditions to calculate the

biliary excretion intrinsic clearance, which leads to variability. We have added these points to the discussion section.

Page 8, Lines 260-272, Discussion

Although SCHs have the merit of maintaining the expression of drug-metabolizing enzymes and transporters for a long enough time to study drug disposition, they have several drawbacks in evaluating hepatobiliary drug disposition. For example, when the excreted drugs accumulate in the closed lumen, a rapid equilibrium results between the cells and the lumen. **In fact, the calculated BEI and *in vivo* biliary excretion intrinsic clearance of the test drugs varied with drug treatment time in sandwich-cultured PXB-cells (Fig. 5c, Supplementary Fig. 7). Moreover, the depletion of Ca^{2+} in the culture medium affects cell viability and NTCP activity²⁰.** icHep enables the continuous recovery of a drug and its metabolites as they become secreted into the bile side chamber; therefore, they can overcome the issues associated with SCHs and be applied to the robust screening of the biliary excretion of drug candidates. **On the other hand, *In vitro* biliary intrinsic clearance ($CL_{bile, in}$) was calculated by subtracting the *in vitro* $CL_{bile, app}$ of control cells from that of icHep. This subtraction leads to variance of the calculated biliary excretion capacity. Improvement of formation of open canaliculus is desired to increase the predictivity.**

Q11:

Evaluation of additional human hepatocyte donors may strengthen or change the conclusion related to PXB cells demonstrating higher hemi-lumen formation of bile spaces.

Response:

We thank the reviewer for this comment. We considered that the formation of open-form canaliculus in PXB-cells is comparable to that of primary human hepatocyte. We also conducted experiments using additional PXB-cells donor as responded to comment #3 as above. In hemi-lumen formation experiment using PXB-cells derived from two human liver donors (JFC and HUM181001B), the frequency of hemi-lumen formation was similar in both donors. This result indicates that the contribution of claudin to hepatocyte hemi-lumen formation is not limited to a specific hepatocyte donor. The results of these experiments are described as Figure 2c, and Supplementary Figure 5. In addition, we deleted the description about comparison of PXB-cells and primary human hepatocytes. Please refer those Figures and Figure legends in the revised manuscript.

Figure 2c (Page 18) and Figure legend (Page 19, Line 9-11)

Supplementary Figure 5 (Page 6) and Figure legend (Page 6, Line 1-5)

We have changed the manuscript in Results section as below.

Page 4, Line 126-132, Results:

When the icHep was constructed with PXB-cells (donor: JFC), which are fresh hepatocytes derived from human liver chimeric mice¹⁴, on supports individually coated with each claudin, hemi-lumen formation rates were $44.5 \pm 4.8\%$, $48.9 \pm 11.7\%$, $38.6 \pm 8.1\%$, and $35.9 \pm 6.4\%$ for claudin-1, claudin-2, claudin-3, and claudin-9, respectively (Supplementary Fig. 5). When all these four claudins were coated in a mixture, hemi-lumen formation rates were $49.2 \pm 4.7\%$ (donor: JFC) and $35.8 \pm 6.3\%$ (donor: HUM181001B), respectively. (Supplementary Fig. 5).

Q12:

In the conclusions three hypothetical scenarios are included without experimental data, one is troglitazone sulfate and hepatotoxicity. Ideally, this could be tested as an example molecule in this system and would enable verification of retention of SULT function and confirmation of utility to test inhibition of biliary efflux transporters. Another example is evaluating the metabolite secretion at the apical and basolateral compartment and lastly, the potential investigation of EHC. It should be noted that microbiome play a significant role in EHC specifically in cleavage of phenolic glucuronides thus, intestinal combo may not be sufficient to assess this potential.

Response:

We thank the reviewer for this comment.

12-1: We mentioned troglitazone as an example of involving drug metabolism for BSEP inhibition. However, we considered that the IC₅₀ values for troglitazone and its sulfate form for BSEP inhibition (2.3 and 4.2 μM , respectively) [Pähler *et al.*, *Advances in Molecular Toxicology*, **2**:25-56 (2008)] were too close and to obtain a clear conclusion. We have previously reported using sandwich-cultured human hepatocytes that BSEP is inhibited by candesartan cilexetil (CIL), but not by its active metabolite candesartan (CAN) [Fukuda *et al.*, *Drug Metab Pharmacokinet*, **29**(1):94-6 (2014)]. Therefore, we decided to conduct additional experiment to show the applicability of icHep as an evaluation system for drug-induced cholestasis including drug metabolism using CIL instead of troglitazone. The results of these experiments are described in Figure 6. Please refer Figure and Figure legend in the revised manuscript. In this experiment, we clearly observed BSEP inhibition by CIL but not by CAN, since esterase inhibitor, which inhibits formation of CAN from CIL, increased accumulation of taurocholate in the cells and decreased biliary excretion clearance of taurocholate. So, we deleted discussion about troglitazone and replaced with the result of CAN. Furthermore, we revised manuscript according to the comment.

Figure 6 (Page 29-30) and Figure legend (Page 30, Line 26-36)

We have changed the manuscript in Results and Discussion section as below.

Page 7, Line 229-246, Results:

Evaluation of drug-induced cholestasis (DIC) involving metabolic processes using icHep.

Inhibition of BSEP by drugs and/or their metabolites can lead to DIC with intracellular accumulation of bile acids and subsequent cholestasis, leading to severe liver injury. Since DIC is often associated with drug metabolism processes, it is desirable to evaluate using SCHs that maintain liver physiology. We have previously reported using SCHs that BSEP is inhibited by candesartan cilexetil (CIL), but not by its active metabolite candesartan (CAN)¹⁸. Here, we investigated the applicability of icHep as an evaluation system for DIC using CIL. Treatment of the cells with CIL alone did not affect biliary excretion clearance of BSEP substrate taurocholate (Fig. 6a). This is due to the rapid hydrolysis of CIL to CAN by esterase in hepatocytes. On the other hand, treatment of the cells with CIL in the presence of esterase inhibitor diisopropyl fluorophosphate (DFP) resulted in the decreased biliary excretion clearance of taurocholate, accompanied by increased intracellular accumulation of taurocholate (Fig. 6a, b). Furthermore, intracellular CIL and CAN were increased and decreased, respectively, by DFP treatment in a concentration dependent manner (Fig. 6c). A significant correlation was observed between the decrease in biliary excretion clearance of taurocholate and the increase in intracellular CIL concentration ($r = -0.984$, $p = 0.01$), whereas there was no correlation with intracellular CAN concentration ($r = 0.274$, $p = 0.66$) (Fig. 6d, e). These results indicate that the icHep permeation assay system can be useful to evaluate DIC, including intrahepatic drug metabolism.

Page 9, Line 290-297, Discussion:

Therefore, we examined the applicability of icHep to the evaluation of biliary excretion inhibition which is affected by intracellular drug metabolism process, using CIL, whose BSEP inhibitory efficacy is attenuated by intracellular metabolism to CAN by esterase. Co-exposure of DFP and CIL decreased biliary clearance of taurocholate with increasing intracellular concentration of CIL. This observation is explained by the inhibition of BSEP-mediated efflux of taurocholate due to decreased intracellular CAN by DFP. Therefore, icHep is applicable for assessment of DIC, including intrahepatic drug metabolism using permeation assays.

12-2: Regarding the assessment of metabolite secretion in the apical and basal compartments, it is difficult at this time to adequately predict substrate distribution in the liver because of concerns about the flux of substrate accumulated in several closed bile ducts in icHep to the sinusoidal medium. Therefore, our claim may have been overstated. To clarify, we have changed the manuscript in Discussion section.

Page 9, Line 305-307, Discussion:

icHep provides easy access to secreted metabolites into the blood and bile side chambers. Therefore, icHep may be used to predict blood levels of metabolites.

12-3: Finally, construction of MPS capable of loading icHep and intestinal cells with microbiome is not currently available, and potential investigation of enterohepatic circulation is a future challenge. We added description about intestinal microbiome.

Page 9, Line 307-313, Discussion:

In addition, biliary-excreted parental drugs and metabolites are reabsorbed in the gastrointestinal tract, in a process called enterohepatic circulation. It has a significant impact on drug pharmacokinetics, efficacy, and safety. MPS is useful for studying inter-organ relationships; however, enterohepatic circulation is difficult to establish using currently available cell culture methods. It is expected that MPS, including organ-on-a-chip systems equipped with icHep, intestinal cells, and intestinal microbiome will enable to evaluate complicated enterohepatic circulation.

Reviewer #2:

We wish to express our appreciation to the reviewer for your insightful comments on our paper. We responded to your comments as followings:

Major comments

Q1:

According to Fig.2d, it seems that coating with a mixture of claudins (-1, -2, -3, -9) did not contribute to more hemi-lumen formation in the PXB cells, compared to coating with single claudins. Please explain it. Why didn't the authors choose using single claudins? The selection of claudins was based on the experiments on HepG2 cells that have lower expression of most claudins, did the current claudins combination is the optimized one to induce bile canaliculus formation in different hepatocyte?

Response:

We thank the reviewer for this comment. Claudins can trans-interact with heterologous claudin molecules expressed on neighboring cell membranes [Furuse *et al.*, *J Cell Biol*, **147**(4):891-903 (1999)]. We used claudin-1, -2, -3, and -9 mixture rather than single claudin because it means that the more diverse the claudin molecules used in the coating, the higher the probability of device-cell claudin interactions. As pointed out by the reviewer, it may be possible to construct an evaluation system with higher functionality by screening the optimal claudin molecules and combinations for human hepatocytes. However, the most important point of our study is to show that the bile secretion process can be evaluated by permeation assay using claudin-coated plate. We would like to set studies of claudin optimization as future work. In response to reviewer's comment, the manuscript was modified as followings.

Figure 1b (Page 16) and Figure legend (Page 17, Line 7-11)

Supplementary Figure 5 (Page 6) and Figure legend (Page 6, Line 1-5)

We added the following sentences to the Discussion section.

Page 8, Line 275-279, Discussion:

On the other hand, claudin was narrowed down using HepG2 cells (Fig. 1c). Since claudin genes expressed in HepG2 cells have different patterns from human hepatocytes (Fig. 1b), it is possible to construct icHep with higher formation of open-form canaliculus by further optimizing the types and combinations of claudin proteins using human hepatocytes.

Q2:

The resolution of MRP2 staining image in Fig.1d is quite low that the bile canaliculi could not be clearly seen. Please supply images with better resolution. In addition, since the formation of bile canaliculi in icHep is critical to whole work, the author could consider more data to support this, for example, the CDFDA staining, because HepG2 cells alone could expressed MRP2 in some extent.

Response:

We appreciate the reviewer's comment on this point. According to the comment, we newly acquired a high-resolution bile canaliculus image.

Figure 1d (Page 17) and Figure legend (Page 17, Line 16-20)

We used HepG2 cells in Figure 1d for initial confirmation of the concept that bile canaliculus localization is induced in the direction of claudin action. On the other hand, the bile canaliculus of HepG2 cells exhibited an immature spherical morphology, making them not suitable tools for assessing the formation of open hemi-lumens. Therefore, the formation of open hemi-lumens by claudin coating was evaluated by MRP2 immunostaining using primary human hepatocytes (Figure 2b) and PXB-cells (Figure 3f) that form more physiological bile canaliculus. In addition, a CDFDA permeation assay using icHep was performed, and the MRP2-mediated transport of CDF to the biliary side (lower chamber) in icHep was significantly higher than that in PXB-cells cultured on a collagen-coated permeable support (Figure 4c). These result supports the biliary excretion of CDF through the open hemi-lumen in icHep.

Figure 2b (Page 18) and Figure legend (Page 19, Line 3-9)

Figure 3f (Page 21) and Figure legend (Page 23, Line 3-6)

Figure 4c (Page 24) and Figure legend (Page 25, Line 7-12)

Q3:

The advance of icHep in pharmacokinetic was mostly characterized by comparing with hepatocyte culture in collagen-coated support. From this perspective, icHep should compare with SCH directly

rather than current control (collagen, no claudins) (Fig4., Fig5.). In SCH system, drug transporters expression could be maintained for at least 3 days, while the expression levels of transporters in icHep peaked at day 3.

Response:

Thank you for thoughtful comment. According to the comment, we conducted experiments, including gene expression changes of pharmacokinetic proteins (transporters and drug metabolizing enzymes), drug metabolizing enzyme activities, and biliary excretion activities, to compare icHep and sandwich-cultured PXB-cells (SCH) derived from donor JFC. In the current protocol, PXB-cells were obtained seeded in flasks, trypsinized and reseeded onto the permeable support. However, based on advice from PhoenixBio, the supplier of PXB-cells, we were concerned about the effect of trypsin treatment on cellular gene expression changes. Therefore, when we implemented a modified protocol in which freshly isolated PXB-cells were seeded on a permeable support, gene expression of transporters and drug-metabolizing enzymes was relatively maintained for 3, 5, 7, and 14 days of culture. Gene expression levels for transporters and drug metabolizing enzymes were comparable between icHep and SCH (Fig. 3d and 3e). Drug metabolizing activity was also comparable between icHep and SCH, while some difference was observed between donors (Fig. 3g and 3h). In addition, effect of transporter inhibitors on permeability of drugs (between Fig. 5a and 5b) and predicted *in vivo* $CL_{bile,int}$ (between Fig. 5d and 5e) were comparable between used two Lot of hepatocytes. Predicted *in vivo* $CL_{bile,int}$ values by BEI was variable depending on incubated time (Fig. 5c and Supplementary Fig. 7).

Figure 3d and 3e (Page 20-21) and Figure legend (Page 22, Line 26-Page 23, Line 3)

Figure 3g and 3h (Page 22) and Figure legend (Page 23, Line 6-15)

Figure 5a-e (Page 26-28) and Figure legend (Page 28, Line 17-31)

Supplementary Figure 7 (Page 8-9) and Figure legend (Page 9, Line 13-19)

We have changed the manuscript in Results section as below.

Page 5, Line 151-156, Results:

Next, gene expression changes of typical hepatic transporters and drug-metabolizing enzymes during the culture period were evaluated. Most of the investigated gene expression was relatively maintained during the culture period, while gene expression levels of some apical (BSEP), and basal (OATP1B1, OATP1B3) transporters and drug-metabolizing enzymes (CYP1A2, CYP2E1) declined by day 3 of culture. Each gene expression profile was comparable between icHep and conventional sandwich culture system (Fig 3d, 3e).

Page 5, Line 161-168, Results:

Furthermore, the activity of drug-metabolizing enzymes in icHep and sandwich-cultured PXB-cells obtained through the cocktail substrate method (see Supplementary Table 1) were measured for major CYPs (CYP1A2, CYP2B6, CYP2C9, CYP2C19, CYP2D6, CYP2E1, CYP3A4), and UGT1A1. icHep

and sandwich-cultured PXB-cells derived from the same donor exhibited similar metabolic activities, consistent with their gene expression profiles (Fig. 3g). Moreover, donor-to-donor differences (JFC and HUM181001B) were identified in the activities of several drug-metabolizing enzymes (CYP1A2, CYP2C9, CYP2E1, CYP3A4, and UGT1A1) in icHep (Fig. 3g).

Page 6, Line 207-Page 7, Line 227, Results:

We studied the predictability of human *in vivo* biliary excretion using drug permeation assays with icHep from two donors and compared them with conventional sandwich-cultured PXB-cells. Seven drugs whose human *in vivo* biliary excretion clearances are known (cimetidine, digoxin, erythromycin, methotrexate, nafcillin, SN-38, and vincristine) were evaluated. In the permeation assays using icHep, the permeability coefficient of each drug was significantly increased compared to that of the control and decreased in the presence of their respective transporter inhibitors (Fig. 4d, 5a, b, Supplementary Fig. 6). Subsequently, the *in vivo* biliary excretion clearance, plasma protein-unbound fraction, and blood/plasma ratio of each drug were obtained from the literature, and their *in vivo* biliary excretion intrinsic clearances in human were estimated using Equation (4) (see Methods section). The pharmacokinetic and biliary excretion parameters of each drug are summarised in Table 1. The predicted human *in vivo* biliary excretion intrinsic clearance of the drugs in icHep exhibited a good correlation with the corresponding human *in vivo* values [icHep (JFC): $r = 0.969$ and $p = 0.001$, icHep (HUM181001B): $r = 0.966$ and $p = 0.001$] (Fig. 5d, e). Furthermore, the BEI of each drug was calculated as an index of the drug biliary excretion activity using sandwich-cultured PXB-cells for comparison with icHep. However, the calculated BEIs were variable depending on the incubation time of each drug to PXB-cells, and did not show a constant value (Fig. 5c). Furthermore, the prediction accuracy of *in vivo* clearance estimated from BEI in PXB-cells differed depending on the incubation time of each drug (2 min incubation: $r = 0.498$ and $p = 0.255$, 5 min incubation: $r = 0.994$ and $p = 0.001$, 10 min incubation: $r = 0.952$ and $p = 0.001$, 20 min incubation: $r = 0.952$ and $p = 0.001$) (Supplementary Fig. 7). Thus, our icHep permeation assay system efficiently enables more reliable prediction of the human *in vivo* biliary excretion of drugs.

Q4:

The author should add the prediction results of human *in vivo* biliary excretion intrinsic clearance estimated from the control (collagen, no claudins) in Table 1. From that, we could clearly know the advantage of this novel system in predicting biliary clearance.

Response:

Thank you for the comment. The *in vitro* biliary excretion intrinsic clearance of icHep was calculated by subtracting that of control from the apparent biliary excretion clearance of icHep obtained by permeation assays. Therefore, *in vitro* and *in vivo* biliary excretion intrinsic clearance from control conditions could not be calculated separately. Instead, we conducted the conventional SCH method to

predict bile secretion to compare with icHep. We considered that icHep will enable stable evaluation, because conventional SCH method yields different results depending on the time point. This point has been described in the original manuscript as following:

Page 7, Lines 219-227, Results:

Furthermore, the BEI of each drug was calculated as an index of the drug biliary excretion activity using sandwich-cultured PXB-cells for comparison with icHep. However, the calculated BEIs were variable depending on the incubation time of each drug to PXB-cells, and did not show a constant value (Fig. 5c). In addition, the prediction accuracy of *in vivo* clearance estimated from BEI in PXB-cells differed depending on the incubation time of each drug (2 min incubation: $r = 0.498$ and $p = 0.255$, 5 min incubation: $r = 0.994$ and $p = 0.001$, 10 min incubation: $r = 0.952$ and $p = 0.001$, 20 min incubation: $r = 0.952$ and $p = 0.001$) (Supplementary Fig. 7). Thus, our icHep permeation assay system efficiently enables more reliable prediction of the human *in vivo* biliary excretion of drugs.

Page 36, Line 6-7, Methods:

In vitro biliary intrinsic clearance ($CL_{bile,int}$) was calculated by subtracting the *in vitro* $CL_{bile,app}$ of control cells from that of icHep to evaluate the true biliary excretion capacity.

Q5:

According to Fig. 5a, the permeability coefficient of nafcillin and SN-38 didn't change after the treatment of inhibitors in icHep control group. Please explain it. In other words, there was no difference in control after inhibitor treatment while huge difference in icHep. Why? Does that come from transporter expression? Did the author measure mRNA or protein levels to verify it?

Response:

Thank you for the comment. PXB-cells cultured on a collagen-coated permeable support (control condition) form closed bile canaliculus between adjacent hepatocytes. Thus, biliary excreted drug accumulates in the closed bile canaliculus, resulting in no biliary transporter-mediated transport of substrate to the lower chamber (bile side). Accordingly, the absence of change in substrate permeability by transporter inhibitor treatment is the expected result.

Q6:

According to Fig. 5b, the $CL_{bile,int}$ of Nafcillin and SN-38 contributed most to the correlation coefficient ($r = 0.998$). The beneath mechanisms should be investigated. Why the compounds in Fig 4 were not included in Fig. 5b, e.g. digoxin, E217 β G and taurocholate?

Response:

We appreciate the reviewer's comment. We understand the comment by the reviewer but this is due to the very limited previous reports on the *in vivo* biliary excretion clearance in human. Among the compounds investigated in Figure 4, digoxin, E₂17βG, and taurocholate, we found the information on human *in vivo* biliary excretion clearance only for digoxin. Therefore, the predicted human *in vivo* biliary excretion intrinsic clearance of digoxin was calculated and subjected to correlation analysis with observed values in human (Figure 5d, 5e, Supplementary Figure 7, and Table 1). On the other hand, human *in vivo* clearance information for E₂17βG and taurocholate was not found as far as we investigated, so they were not included in the correlation diagram.

We understand that the intrinsic biliary excretion clearance of SN-38 and Nafcillin contributed most to the correlation coefficient. We calculated the intrinsic clearance in human *in vivo* biliary excretion according to Equation (4) in the Methods section. Due to the presence of the plasma unbound fraction (f_p) term in the denominator of Equation (4), intrinsic clearance values are significantly affected by protein binding of the test compound. The plasma unbound fraction, f_p values of SN-38 and Nafcillin are 0.00887 and 0.123, respectively (Table 1). This value is the lowest and second lowest among the tested compounds. Therefore, the high intrinsic clearance values of SN-38 and Nafcillin may be attributed to their high protein binding properties.

Figure 5d-e (Page 27-28) and Figure legend (Page 28, Line 28-29)

Supplementary Figure 7 (Page 8-9) and Figure legend (Page 9, Line 13-19)

Table 1 (Page 11)

Q7:

The author should indicate the ratio of human-derived hepatocyte in PXB cells, as we know hepatocytes from human liver chimeric mice contain both mouse and human cells. This is important to the correlation of biliary clearance between icHep and human results.

Response:

Thank you for thoughtful comment. Microscopic observation of hepatocytes treated with 66Z-conjugated magnetic beads was performed to confirm the ratio of human hepatocytes in PXB-cells and confirmed that the ratio of human hepatocytes was $90.3 \pm 2.9\%$ (22 animals) [Yamasaki *et al.*, *PLOS One*, **15**(9):e0237809 (2020)]. This indicates that about 10% of mouse hepatocytes are present in PXB-cells, but almost all hepatocytes are replaced with human type.

We have changed the manuscript in Discussion and Methods section as below.

Page 8, Line 272-275, Discussion:

In addition, PXB-cells used for constructing icHep contain about 10% mouse-derived cells²¹. Therefore, some of the obtained biliary transport includes mouse-derived reactions. Further study is

required to establish icHep with pure human hepatocytes such as improved primary cultured hepatocytes^{22,23} and iPS-derived hepatocytes²⁴.

Page 33, Line 33-35, Methods:

Microscopic observation of hepatocytes treated with 66Z-conjugated magnetic beads revealed an average ratio of human hepatocytes to total freshly isolated PXB-cells of $90.3 \pm 2.9\%$ (22 animals)²¹.

Minor comments

Q1:

Fig. 1c, please provide the mRNA or protein expression results of claudins in HepG2 cells after transfected with claudin-expressing plasmids separately.

Response:

Thank you for thoughtful comment. As pointed out by the reviewer, claudin mRNA expression analysis was performed after individual transfection of claudin plasmids. Gene expression of each claudin increased in the plasmid-introduced group compared to the control group (Supplementary Figure 2).

Supplementary Figure 2 (Page 3) and Figure legend (Page 3, Line 1-5)

We have changed the manuscript in Results section as below.

Page 3, Line 94-95, Results:

All claudin molecules investigated were efficiently transfected into HepG2 cells by the lentiviral method (Supplementary Fig. 2).

Q2:

Figure1b, consider creating a segment (break) to Y axis to better compare the expression of claudin-4, -11, -19 between two cells.

Response:

Thank you for your comments. Separate graphs have been added for some claudin molecules (CLDN8, 17, 18, 19, 25 and 27) to facilitate comparison of claudin expression between the HepG2 cells and human liver in Fig 1b.

Figure 1b (Page 16) and Figure legend (Page 17, Line 7-11)

Q3:

Page3 Line96, Page6 Line188, please change “increase by” to “increase to”.

Response:

Thank you for your comments. The relevant parts of the manuscript have been corrected as appropriate.

Page 3, Line 95-97, Results:

The expression of claudin-1, -2, -3, and -9 **increased** the number of bile canaliculi in HepG2 cells to 150 ± 14 , 158 ± 15 , 145 ± 21 , and $142 \pm 15\%$, respectively, compared to mock cells (Fig. 1c).

Page 6, Line 193-195, Results:

When CDFDA was loaded onto the sinusoidal side (upper chamber), the appearance of CDF on the bile side (lower chamber) of icHep **increased to** $147 \pm 14.6\%$ of the control group.

Q4:

Fig. 3, the figure title is inappropriate since it was the drug uptake assay instead of drug permeation assay.

Response:

Thank you for your comments. Figure 3 legend title changed to: **Pharmacokinetic functional evaluation of icHep.**